# SAS: Simulated Attention Score

Chuanyang Zheng[1][*]   Jiankai Sun[2,3]   Yihang Gao[4]   Yuehao Wang[5]   Peihao Wang[5]
Jing Xiong[6]   Liliang Ren[3]   Hao Cheng[3]   Janardhan Kulkarni[3]   Yelong Shen[3]
Zhangyang Wang[5]   Mac Schwager[2]   Anderson Schneider[1]   Xiaodong Liu[3]   Jianfeng Gao[3]

[1]Morgan Stanley   [2]Stanford   [3]Microsoft Research   [4]NUS   [5]UT Austin   [6]HKU

## Abstract

The attention mechanism is a core component of the Transformer architecture. Various methods have been developed to compute attention scores, including multi-head attention (MHA), multi-query attention, group-query attention and so on. We further analyze the MHA and observe that its performance improves as the number of attention heads increases, provided the hidden size per head remains sufficiently large. Therefore, increasing both the head count and hidden size per head with minimal parameter overhead can lead to significant performance gains at a low cost. Motivated by this insight, we introduce Simulated Attention Score (SAS), which **maintains a compact model size while simulating a larger number of attention heads and hidden feature dimension per head.** This is achieved by projecting a low-dimensional head representation into a higher-dimensional space, effectively increasing attention capacity without increasing parameter count. Beyond the head representations, we further extend the simulation approach to feature dimension of the key and query embeddings, enhancing expressiveness by mimicking the behavior of a larger model while preserving the original model size. **To control the parameter cost, we also propose Parameter-Efficient Attention Aggregation (PEAA).** Comprehensive experiments on a variety of datasets and tasks demonstrate the effectiveness of the proposed SAS method, achieving significant improvements over different attention variants.

## 1   Introduction

The Transformer architecture [77] has emerged as the predominant framework in modern machine learning, demonstrating remarkable success across diverse domains. Transformer-based models consistently achieve state-of-the-art performance in numerous natural language processing tasks, including machine translation [5, 89], question answering [96, 27, 4], and commonsense reasoning [71, 106, 99, 92, 56]. The fundamental components of the Transformer comprise attention mechanisms and feed-forward networks, with attention score computation serving as a critical element. Beyond the original multi-head attention (MHA) formulation [77], researchers have developed various efficient sparse attention variants, such as sliding window approaches (e.g., Streaming LLMs [80]), linear transformers (e.g., Performer [14]), and sparse attention mechanisms (e.g., Reformer and Sparse Sinkhorn transformer [73, 21]).

Modern architectures employ diverse strategies for attention score computation. The standard MHA approach [77] maintains identical head counts across query, key, and value embeddings. To optimize memory efficiency, subsequent innovations have introduced parameter sharing schemes: Multi-Query Attention (MQA) [61] and Group-Query Attention (GQA) [3] share key and value projections across attention heads. The Multi-Latent Attention (MLA) approach [40] adopts low-rank compression

---

[*]Contact Email: cyzhengme@gmail.com

39th Conference on Neural Information Processing Systems (NeurIPS 2025).

for key and value projections, caching only latent representations. More recently, Tensor Product Attention (TPA) [100] has introduced tensor decomposition techniques for compact representation of queries, keys, and values. We further analyze the MHA in Figure 1, and we find that the MHA performance gradually increases when the attention head number increases, as long as the hidden size per head is not too small. Hence, by increasing the number of attention heads and the hidden size per head at a minimal cost in terms of additional parameters, we can expect a substantial improvement in performance.

Building on these observations and insights, we propose Simulated Attention Score (SAS) to enhance Transformer performance through simulated attention head and feature expansion. Traditional query embeddings have dimensionality $[B, T, H, D]$, where $B$ represents the batch size, $T$ is the sequence length, $H$ denotes the head count, and $D$ is the feature dimension per head. Our approach applies linear projections with nonlinear activations along the head dimension ($H$) to effectively increase the number of attention heads to $\hat{H}$ with $\hat{H} > H$. Similarly, we also simulate more feature dimensions ($D$) for query/key embedding for the attention score computation. The proposed method effectively and efficiently expands both the head and feature dimensions, resulting in a query representation of shape $[B, T, \hat{H}, \hat{D}]$, where the expanded dimensions $\hat{H}$ and $\hat{D}$ exceed the original dimensions $H$ and $D$, respectively. This expansion simulates a greater number of attention heads and a higher hidden feature dimensionality per head, thereby enhancing model expressiveness while preserving a compact overall model size. Additionally, we propose the parameter-efficient attention aggregation to control parameter size and computation. To summarize, our main contributions are the following:

1. By analyzing previous attention variants, we empirically observe that the number of attention heads plays a critical role in the Transformer performance, while more attention heads can lead to better performance, as long as the hidden size per head is not too small.

2. Based on the above observation, we propose SAS to use a moderately sized model to simulate the attention mechanism of a larger model, effectively increasing attention heads and hidden dimension per head. To control the parameter cost, we also propose Parameter-Efficient Attention Aggregation (PEAA).

3. We conduct extensive experiments on various datasets, tasks, and hyperparameter settings to validate the effectiveness of the proposed method, under different training lengths, model sizes, and datasets. Furthermore, we pretrain the model with SAS on the large-scale datasets and evaluate its performance on a wide range of downstream tasks, further showing the effectiveness of the SAS method.

## 2   Related Works

**Attention Mechanism in Transformer.**   The Transformer architecture, leveraging Multi-Head Attention (MHA) [77] for effective long-range dependency modeling [84, 85], suffers from high memory footprint and bandwidth demands during training. To alleviate these issues, several efficient attention mechanisms have been proposed. Multi-Query Attention (MQA) [61] reduces memory usage by sharing key and value heads among multiple queries. Grouped-Query Attention (GQA) [3] addresses MQA's drawbacks by grouping queries with shared key and value heads. Multi-Head Latent Attention (MLA) [40] optimizes inference through low-rank key and value representations, surpassing MHA with a smaller KV cache. Multi-matrix Factorization Attention (MFA) [33] builds upon MQA by factorizing the query matrix. Tensor Product Attention (TPA) [100] achieves performance improvements and KV cache reduction by employing contextual tensor decomposition for query, key, and value representations.

**Linear Attention and RNN.**   Linear recurrent language models have emerged as a compelling alternative due to their training and inference efficiency. The evolution of these models has progressed from early linear RNNs with data-independent decay, exemplified by S4 [26], S5 [65], LRU [48], RWKV4/5 [51, 52], and RetNet [70], towards more sophisticated architectures incorporating data-dependent decay, such as HGRN [55], Mamba [25, 19], and GSA [101]. This transition highlights the effectiveness of gating and forgetting mechanisms (termed "selective mechanisms" in Mamba), a principle inspired by classical gated RNNs. A key distinction in modern forget gates, unlike those in LSTMs [24], is their independence from the previous hidden state, relying solely on input data and enabling efficient parallelization [55]. Further advancements include Longhorn [41] and

TTT [69], which reformulate state space learning as a gradient-based online optimization, and Gated DeltaNet [91], which enhances a linear RNN's expressiveness through gating while preserving training efficiency. [63] provides a comprehensive comparison of expressivity between Attention-based architectures, Linear attention mechanisms, and Recurrent Neural Networks.

**Non-Linear Mapping.** The non-linear mappings typically have a higher expressiveness than linear mappings. Neural networks are well-suited for implementing such non-linear transformations. The most basic neural network architecture for this purpose is the multilayer perceptron (MLP) [53], which is composed of linear transformations and non-linear activation functions. In certain data structures and configurations, the MLP can be replaced with a convolutional neural network (CNN) [36, 35] to better capture spatial or local patterns. Common activation functions include ReLU[2], Sigmoid [29], Tanh, Leaky ReLU [88], among others.

## 3 Method

### 3.1 The Number of Attention Heads Plays an Important Role

To investigate the impact of varying head counts on model performance, we conduct experiments with MHA models using different numbers of attention heads, ranging from 1 head up to 96 heads, alongside a hidden feature of 768 dimensions. The model with 12 heads contains 125 million parameters. The results of these experiments are illustrated in Figure 1. When only a single attention head (1 head) is used, the model achieves a perplexity score of 6.08. However, as the number of attention heads incrementally increases, the model's performance improves, reaching an optimal perplexity of 5.82 when the head count matches the original configuration

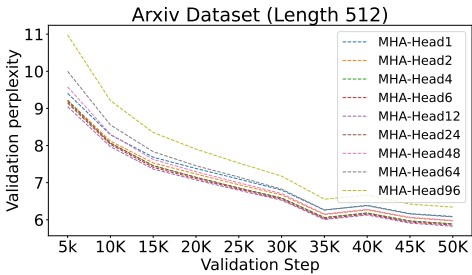

Figure 1: The performance of MHA with different head numbers on the Arxiv dataset, with a 125M model (hidden size is 768) and training length 512.

of 12 heads. Beyond this point, further increasing the number of attention heads leads to a gradual degradation in performance, as evidenced by the rising perplexity scores. Therefore, more attention heads and larger hidden sizes per head may further improve the performance.

### 3.2 Simulated Attention Score

Inspired by the above observations, we propose Simulated Attention Score (SAS), a method that simulates the behavior of a larger Transformer model with more attention heads and larger hidden size per head, using a smaller and more efficient model. Our approach maintains a moderate model size while significantly enhancing representational expressiveness. The implementation is presented in Appendix L.

**Initial query, key and value embeddings.** Theoretically, our method is compatible with various existing attention mechanisms, including MQA, GQA, MHA, and TPA (e.g. the result presented in Appendix J), to construct the initial query and key tensors. Among these, MHA offers relatively stable performance, and it is widely used in previous works [77, 34]. Therefore, we adopt MHA as the foundation for initializing the query and key embeddings in our design.

**Expanding feature dimensions enhances model performance.** In addition to simulating more attention heads, we extend our method to include feature dimension expansion, as the feature dimension also plays a critical role in shaping attention representations. We apply non-linear mapping to both the query and key tensors to simulate a larger feature space. As a result, SAS enhances attention modeling through a dual expansion, which increases both the number of heads and the feature dimension. Considering computational efficiency, we apply feature dimension expansion only to the key and query tensors, leaving the value tensor with its unchanged feature dimension.

### 3.3 Simulating Attention Heads via Non-linear Mappings

In this section, we present the step-by-step procedure for simulating a greater number of attention heads using a non-linear mapping. Our goal is to enrich the model's expressiveness by mimicking the behavior of a wider multi-head attention mechanism, while keeping the actual number of parameters and computational cost moderate.

Given an input query tensor $\mathbf{Q} \in \mathbb{R}^{B \times T \times H \times D}$, where $B$ denotes the batch size, $T$ is the sequence length, $H$ represents the number of attention heads, and $D$ is the feature dimension per head. This tensor structure follows the standard formulation in multi-head attention, where each head processes a D-dimensional feature vector independently.

We first reshape the query tensor to combine the batch size, sequence length, and feature dimensions, isolating the head dimension for further processing:

$$\mathbf{Q}_0 = \text{Reshape}(\mathbf{Q}, (B \cdot T \cdot D, H)) \in \mathbb{R}^{(B \cdot T \cdot D) \times H}. \tag{1}$$

Our goal is to simulate a larger number of attention heads to enhance the model's expressiveness. To achieve this, we apply transformations along the head dimension. This reshaping step is crucial, as it allows us to treat the head dimension as the primary axis for non-linear mapping while collapsing the other dimensions into a single batch-like dimension.

The transformation applied to the head dimension consists of a (two-layer) multilayer perceptron (MLP):

$$\mathbf{Q}_1 = \text{Linear}_1^{Q_h}(\mathbf{Q}_0), \tag{2}$$
$$\mathbf{Q}_2 = \text{ReLU}(\mathbf{Q}_1), \tag{3}$$
$$\hat{\mathbf{Q}} = \text{Linear}_2^{Q_h}(\mathbf{Q}_2) + \mathbf{Q}_1 \tag{4}$$

where $\text{Linear}_1^{Q_h} : \mathbb{R}^H \to \mathbb{R}^{\hat{H}}$ and $\text{Linear}_2^{Q_h} : \mathbb{R}^{\hat{H}} \to \mathbb{R}^{\hat{H}}$ are general linear transformation layers. Here, $H$ denotes the original number of attention heads, and $\hat{H}$ represents the expanded (simulated) head dimension. To improve training stability and mitigate the risk of vanishing gradients introduced by the transformation, we incorporate a residual (skip) connection from $\mathbf{Q}_1$ to $\hat{\mathbf{Q}}$. The additional parameter cost introduced by this transformation is negligible, as both the original head dimension $H$ and the expanded head dimension $\hat{H}$ are significantly smaller than the feature dimension $D$. Consequently, in auto-regressive Transformer models, the majority of the parameters are concentrated in the processing of feature vectors rather than in the introduced transformations along head dimensions.

In addition to MLP-based transformations, convolution is another widely adopted approach for dimensional expansion, particularly in computer vision tasks. In our setting, we treat the head dimension as the channel and the feature dimension as a structured 1D representation, analogous to multi-channel feature maps. This interpretation allows us to apply convolutional operations effectively. This perspective enables us to exploit local patterns across attention heads and features, introducing an alternative mechanism for enhancing representational capacity with feature-aware processing of heads. We begin by reshaping the query tensor as follows:

$$\mathbf{Q}_0 = \text{Reshape}(\mathbf{Q}, (B \cdot T, H, D)) \in \mathbb{R}^{(B \cdot T) \times H \times D}.$$

This reshaping step results in a stack of $B \cdot T$ multi-channel features, each of shape $H \times D$, corresponding to individual time steps across the batch. We can then apply convolutional layers with ReLU non-linearity to expand the representational space of these features, enabling more expressive modeling of attention head interactions. The simulation of an expanded head dimension is performed as follows:

$$\mathbf{Q}_1 = \text{Conv}_1^{Q_h}(\mathbf{Q}_0), \tag{5}$$
$$\mathbf{Q}_2 = \text{ReLU}(\mathbf{Q}_1), \tag{6}$$
$$\hat{\mathbf{Q}} = \text{Conv}_2^{Q_h}(\mathbf{Q}_2) + \mathbf{Q}_1, \tag{7}$$

where $\text{Conv}_1^{Q_h} : \mathbb{R}^{H \times D} \to \mathbb{R}^{\hat{H} \times D}$ and $\text{Conv}_2^{Q_h} : \mathbb{R}^{\hat{H} \times D} \to \mathbb{R}^{\hat{H} \times D}$ are standard convolution layers. Here, $H$ denotes the original number of attention heads (interpreted as channels) and $\hat{H}$ is the expanded (simulated) head dimension. Similarly, we introduce the ReLU nonlinear activation function

and the residual connection to enhance representational capacity and improve training stability. The channel expansion is efficiently implemented using convolutional operators. Moreover, convolution naturally facilitates feature sharing across different heads, promoting effective communication both spatially and temporally (i.e., across channels). Note that the convolution operation with a kernel size of one is equivalent to the MLP applied along the channel dimension. Therefore, in our implementation, we adopt convolutional operators for head dimension expansion due to their efficiency and flexibility.

**Head expansion for query tensor Q, key tensor K, and value tensor V.** In the Transformer architecture, each attention head operates on a distinct query, key, and value tensor. Accordingly, the head expansion technique described above is applied to the query ($\mathbf{Q}$), key ($\mathbf{K}$), and value tensors ($\mathbf{V}$), using the same expanded head dimension $\hat{H}$. The additional parameters and computational overhead introduced by this head expansion mechanism are minimal, as both the original and expanded head dimensions ($H$ and $\hat{H}$) are significantly smaller than the feature dimension $D$. As a result, the overall parameter size and computational cost remain comparable to the original Transformer computation. A detailed comparison of the parameter sizes between the vanilla Transformer and our proposed head-simulation method is provided in the experimental section.

### 3.4 Simulating Features via Non-linear Mappings

In addition to the number of attention heads, the feature dimension also plays a critical role in determining the capacity of Transformer models. Building upon the concept of attention head simulation, we extend the idea to simulate larger feature dimensions, aiming to enhance model expressiveness while preserving a moderate model size and computational cost. After performing head dimension simulation, we apply a similar technique to the feature dimension. Taking the query tensor $\mathbf{Q}$ as an example, we first reshape the tensor to isolate the feature dimension for processing:

$$\mathbf{Q}_0 = \text{Reshape}(\mathbf{Q}, (B \cdot T \cdot H, D)), \tag{8}$$

$$\mathbf{Q}_1 = \text{Linear}_1^{Q_f}(\mathbf{Q}_0), \tag{9}$$

$$\mathbf{Q}_2 = \text{ReLU}(\mathbf{Q}_1), \tag{10}$$

$$\hat{\mathbf{Q}} = \text{Linear}_2^{Q_f}(\mathbf{Q}_2) + \mathbf{Q}_1, \tag{11}$$

where $\text{Linear}_1^{Q_f} : \mathbb{R}^D \to \mathbb{R}^{\hat{D}}$ and $\text{Linear}_1^{Q_h} : \mathbb{R}^{\hat{D}} \to \mathbb{R}^{\hat{D}}$ are standard linear transformation layers. Here, $D$ denotes the original feature dimension, and $\hat{D}$ represents the expanded (simulated) feature dimension. This procedure mirrors the head dimension simulation described earlier, with the key difference being that the transformation is now applied along the feature dimension instead of the head dimension. As before, we incorporate a ReLU non-linear activation and a residual connection to improve both representation capacity and training stability.

**Feature dimension expansion for the query tensor Q and key tensor K, excluding the value tensor V.** We apply feature dimension expansion (simulation) to the query tensor $\mathbf{Q}$ and key tensor $\mathbf{K}$, which form the core components of the attention mechanism. However, we do not apply this expansion to the value tensor $\mathbf{V}$. This is because the value tensor typically determines the output token representation, which is subsequently passed through a feedforward MLP layer. Expanding the value dimension would directly increase the token embedding size, thereby leading to a significant increase in the parameter size and computational cost of the feedforward layer. This is undesirable for maintaining model efficiency. Therefore, to balance expressiveness and efficiency, we restrict feature dimension expansion to only $\mathbf{Q}$ and $\mathbf{K}$.

### 3.5 Parameter-Efficient Attention Aggregation (PEAA)

To control the parameter cost, we utilize Parameter-Efficient Attention Aggregation (PEAA). Denote the expanded key, query, and value tensors at $i$-th head by $\hat{\mathbf{K}}_i$, $\hat{\mathbf{Q}}_i$, and $\hat{\mathbf{V}}_i$, respectively. The output from the $i$-th head after the attention is obtained by

$$\mathbf{h}_i = \hat{\mathbf{V}}_i \cdot \text{Softmax}\left(\hat{\mathbf{K}}_i^\top \hat{\mathbf{Q}}_i\right).$$

Then, the attention features are aggregated by

$$\mathbf{Output} = \frac{1}{\hat{H}/H} \sum_{i=1}^{\hat{H}/H} \mathrm{Concat}(\mathbf{h}_{(i-1)\times H+1}, \cdots, \mathbf{h}_{i\times H})\mathbf{W}^O, \tag{12}$$

for $i = 1, 2, \cdots, \hat{H}/H$, where we set $\hat{H}/H$ as a positive integer, and $\mathbf{W}^O \in \mathbb{R}^{(H\cdot D)\times(H\cdot D)}$ is responsible for a linear projection. In general, the attention feature aggregation is conducted by concatenating the output $\mathbf{h}_i$ among all expanded heads followed by a linear projection. However, due to the expanded (simulated) heads, it will introduce additional parameters and computation overhead. Here, we adopt the PEAA method that aggregates the attention features by averaging over groups of heads, as shown in Equation (12).

## 4 Experiment

**Baselines.** We evaluate the proposed SAS against a range of established baselines, including MHA [77], MQA [61], GQA [3], MLA [40] and TPA [100].

**Datasets.** Our analysis involves training language models on the Arxiv and Books3 datasets, which are frequently used benchmarks for evaluating model performance [54, 12, 37, 20]. Also, we train the model on the large-scale dataset FinWeb-Edu [45].

**Experiment settings.** Initially, we compare SAS with other baselines at training lengths 512, and 1024, with model size 125M decoder-only Transformers [7], whose configuration is shown in Appendix C. Subsequently, we evaluate the performance of larger model sizes with 350M and 2.7 B. We also train on large-scale datasets and evaluate downstream tasks.

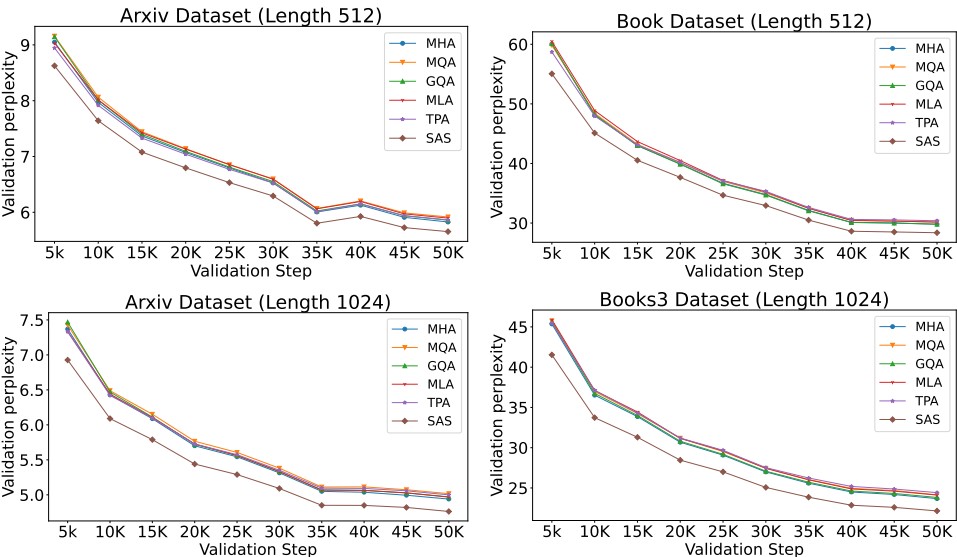

Figure 2: The performance of different methods on the Arxiv and Books3 dataset, with training lengths of 512 and 1024.

### 4.1 Compare SAS with Baselines

**SAS achieves better performance on different training lengths and datasets.** As shown in Figure 2, compared with baselines, SAS achieves the best performance on different training lengths. For example, on the Arxiv dataset with training lengths 512 and 1024, the SAS achieves the lowest perplexity (ppl) 5.65 and 4.76, while the MHA achieves 5.82 and 4.93. Similarly, on the Books3 dataset with training lengths 512 and 1024, SAS achieves 28.37 and 22.14 ppl, while MHA achieves 29.80 and 23.67 ppl. These results suggest that SAS is a robust and effective approach, performing better across different training lengths and datasets.

**SAS saves training token number.**   MHA achieves 29.86 ppl at the final training step (50K iterations) on the Books3 dataset and with a training length 512. SAS already achieves 30.49 ppl at training step 30K. Therefore, SAS could save about 40% training tokens. Similarly, MHA achieves 23.67 ppl at the final training step, with Books3 dataset and training length 1024, SAS achieves 23.85 ppl at training step 30K. We have a similar observation on the Arxiv dataset. In summary, the proposed SAS can achieve certain performance (low ppl) with fewer training steps, thus saving training tokens and computational costs during training.

**SAS maintains better performance on longer training length.**   As shown in Figure 3, SAS achieves the best performance (the lowest 18.34 ppl), while the standard MHA achieves 19.72 ppl. In comparison, other baselines such as GQA, MQA, MLA, and TPA achieve 20.16, 19.59, 19.87 and 20.25 ppl, respectively. These results show that SAS consistently excel even with a longer training length of 2048, indicating its potential to perform well with longer training lengths.

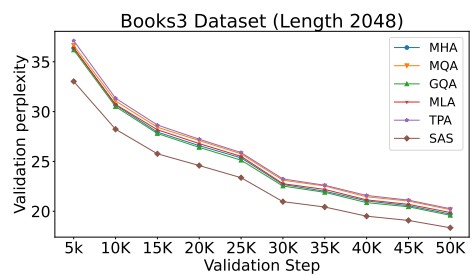

Figure 3: The performance on training length 2048.

## 4.2   The Performance for Larger Models

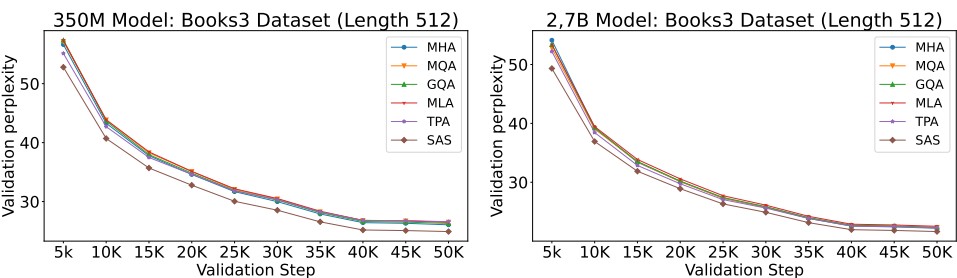

Figure 4: The performance of different methods on the Books3 dataset, with model sizes of 350M and 2.7 B.

**SAS achieves better performance for larger model sizes.**   As shown in Figure 4, when the model size is 350M, SAS achieves 24.91 ppl at the final training step (50K steps), while MHA achieves 26.05 ppl. GQA, MQA, MLA, and TPA achieve 26.50, 26.29, 26.48, and 26.58 ppl. When we consider the model of size 2.7B, SAS achieves 21.62 ppl at the final training step, while MHA achieves 22.18 ppl. GQA, MQA, MLA, and TPA archive 22.29, 22.34, 22.52, and 22.25 ppl, respectively. These results demonstrate SAS's ability to leverage larger model sizes to achieve better performance.

**SAS still achieves better performance across all validation steps, on larger model sizes.**   Across validation sizes ranging from 5K to the final 50K, SAS consistently outperforms competing methods, demonstrating not only superior final performance but also sustained improvements at every stage of training. Notably, SAS achieves this while exhibiting faster convergence, significantly reducing loss earlier in training compared to alternatives, highlighting its superiority in both training efficiency and ultimate model accuracy.

## 4.3   Ablation Study

**The Effect of Head Simulation.**   As shown in Figure 5, the head simulation is employed to increase the attention head number. The SAS with only head simulation and 12 attention heads achieves 23.17 ppl, which is better than MHA with 12 attention heads, achieving 23.67 ppl. This suggests that SAS head simulation further improves the expressiveness, even with the same attention head number. Also, the proposed SAS without head simulation (FeatureOnly) achieves 22.79 ppl, with 36 attention heads, which achieves lower performance than SAS with both head simulation, feature simulation and 36

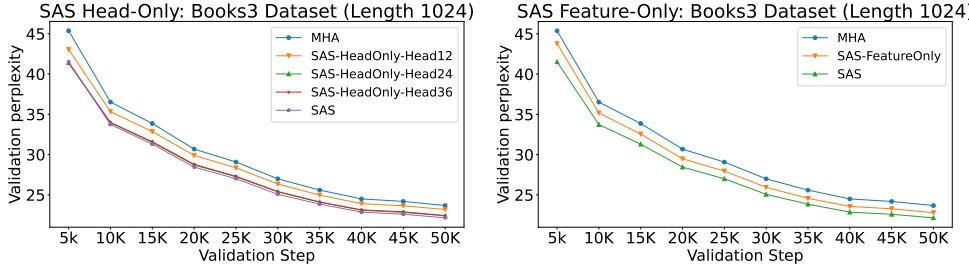

Figure 5: The Ablation study of SAS (SAS-HeadOnly and SAS-FeatureOnly) on the Books3 dataset, with training length 1024.

attention heads, achieving 22.14 ppl. This suggests that the head simulation is important to improve performance.

**The Effect of Feature Simulation.** As shown in Figure 5, the feature simulation is used to increase the query and key feature dimension. Empirical results demonstrate its effectiveness: on the Books3 dataset with a training length of 1024, the SAS-FeatureOnly achieves 22.79 ppl, which is better than MHA with 23.67 ppl. Furthermore, SAS without feature simulation (HeadOnly) achieves a higher perplexity of 22.43, which is worse than SAS with 22.14 ppl. This suggests that the feature simulation is crucial to improve the performance.

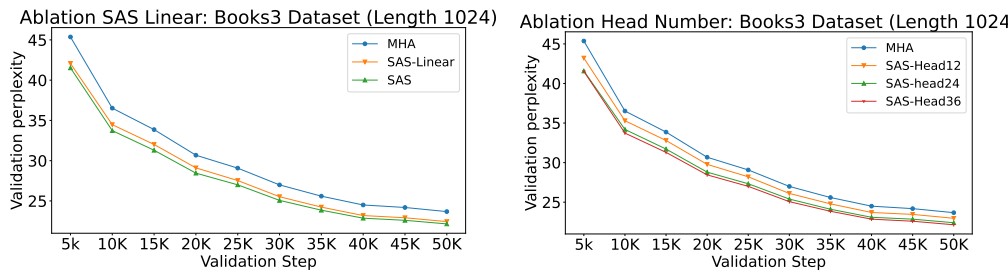

Figure 6: The ablation study of SAS-Linear and different attention head number of SAS, with training length 1024 and Books3 dataset.

**The Effect of Non-Linear Mapping.** SAS adopts ReLU as the nonlinear activation function in the mapping of simulation. As shown in Figure 6, with non-linear mapping, SAS achieves 22.14 perplexity on the Books3 dataset with training length 1024. SAS without non-linear mapping achieves 22.43 ppl, which is higher than SAS with non-linear mapping and lower than MHA. This suggests the advantages of introducing nonlinear mappings in simulation.

**The Effect of Attention Head Number.** As shown in Figure 6, when the attention number is 12, SAS achieves 22.96 ppl, outperforming MHA. This implies that SAS can improve the expressiveness of query, key and value embeddings, leading to better performance even with the same attention head number. Moreover, increasing the attention head number from 12 to 36 leads to further performance gains, with SAS achieving a perplexity of 22.14. This demonstrates that SAS benefits from additional attention heads, which enhance the expressiveness of query, key and value embeddings and result in better performance compared to MHA.

**The Effect of Kernel Size.** As shown in Appendix E with a 125M model, a kernel size of 1 for the head convolution yields significant performance gains. With only less than about 0.4% additional parameters, which is $12 \times ((12 \times 36 + 36 \times 36) \times 3 + (64 \times 96 + 96 \times 96) \times 2) = 0.43M$, SAS is able to significantly improve the performance, reducing the perplexity from 23.67 to 22.50 in MHA. Moreover, as the model size increases, the additional parameter cost ratio becomes even smaller, such as less than 0.1% for a 2.7B model. The results also show that as the kernel size increases from 1 to 7, the perplexity gradually decreases, with the performances of kernel sizes 5 and 7 being comparable.

These findings suggest that SAS can improve performance even with a small kernel size of 1, and the additional parameter cost ratio may gradually decrease as the model size becomes large.

## 4.4 The Performance on Large-Scale Pretrain Dataset

Table 1: Main language modeling results against different methods. All models are trained on the same subset of the FineWeb-Edu dataset [45] with the GPT-2 tokenizer and 100K training steps.

| Model Metric | ARC-E acc_n | ARC-C acc_n | Hellaswag acc_n | PIQA acc_n | ScIQ acc_n | SocialIQA acc | Winograde acc | Avg |
|---|---|---|---|---|---|---|---|---|
| *350M params / 50B tokens* | | | | | | | | |
| MHA | 56.57 | 29.01 | 45.45 | 69.10 | 76.40 | **40.94** | 52.96 | 52.92 |
| MQA | 58.12 | 30.55 | 45.71 | 69.10 | 78.70 | 39.25 | 51.46 | 53.27 |
| GQA | 57.62 | 30.38 | 46.04 | 68.99 | 76.20 | 39.56 | 53.28 | 53.15 |
| MLA | 57.15 | 28.92 | 44.77 | 67.95 | 75.90 | 38.89 | 53.67 | 52.46 |
| TPA | 59.09 | **32.08** | 46.51 | 69.53 | 76.20 | 39.82 | 53.12 | 53.76 |
| SAS | **60.44** | 31.66 | **47.79** | **70.67** | **80.70** | 40.07 | **54.14** | **55.07** |
| *350M params / 10B tokens* | | | | | | | | |
| MHA | 54.29 | 27.99 | 39.94 | 66.38 | 74.00 | 37.67 | 53.59 | 50.55 |
| MQA | 54.45 | 28.75 | 40.51 | 66.21 | 73.10 | 37.92 | 51.30 | 50.32 |
| GQA | 53.03 | 27.73 | 40.34 | 66.59 | 74.20 | 38.95 | 51.22 | 50.29 |
| MLA | 53.28 | 27.65 | 39.51 | 66.10 | **76.00** | 38.08 | 52.09 | 50.39 |
| TPA | 52.44 | 27.99 | 41.27 | 67.57 | 72.60 | 38.33 | 53.35 | 50.51 |
| SAS | **55.93** | **29.61** | **43.04** | **68.82** | 75.90 | **38.43** | **53.20** | **52.13** |
| *760M params / 10B tokens* | | | | | | | | |
| MHA | 56.57 | 29.01 | 45.45 | 67.25 | 77.20 | 38.54 | 52.96 | 52.43 |
| MQA | 55.85 | 29.95 | 43.63 | 67.85 | 76.20 | 39.30 | 53.35 | 52.32 |
| GQA | 54.21 | 29.35 | 44.40 | 68.34 | 77.70 | 38.38 | 52.17 | 52.08 |
| MLA | 57.15 | 29.10 | 44.77 | 67.95 | 75.90 | 38.89 | 53.67 | 52.49 |
| TPA | 58.12 | 30.89 | 44.71 | 69.37 | 77.90 | 39.30 | 52.80 | 53.30 |
| SAS | **59.43** | **31.91** | **45.84** | **69.53** | **78.30** | **39.61** | **55.25** | **54.27** |
| *1.5B params / 10B tokens* | | | | | | | | |
| MHA | 57.87 | 31.40 | 45.51 | 68.82 | 75.90 | 38.18 | 52.33 | 52.86 |
| MQA | 55.85 | 31.74 | 46.40 | 69.53 | 77.30 | 38.13 | 54.70 | 53.38 |
| GQA | 57.62 | 30.20 | 46.20 | 69.48 | 76.30 | 38.95 | 53.59 | 53.19 |
| MLA | 57.24 | 29.95 | 44.90 | 68.50 | 75.20 | 39.76 | 53.43 | 52.71 |
| TPA | 59.81 | 31.23 | 46.84 | 68.99 | 75.60 | 39.46 | **55.09** | 53.86 |
| SAS | **60.44** | **34.39** | **48.66** | **70.08** | **81.40** | 39.92 | 54.93 | **55.69** |

**Downstream Evaluation.** We evaluate zero-shot and two-shot performance on standard benchmarks, including ARC [18], HellaSwag [93], PIQA [6], ScIQ [78], SocialIQA [60] and WinoGrande [59], using the `lm-evaluation-harness` codebase [22]. For ARC-E, ARC-C, HellaSwag, PIQA, and ScIQ, we report accuracy norm; for other tasks, we report standard accuracy. We display the zero-shot evaluation results of models here in Tables 1.

**SAS behaves well from small data size (e.g., 10B) to larger data size (e.g. 50B).** When the training tokens are 10B, the 350M model achieves the best performance on most benchmarks, such as ARC-E, Hellawag and so on. Also, SAS achieves the best average performance of 52.13. When the training tokens number increases from 10B to 50B, SAS still achieves the best performance on most benchmarks, and SAS achieves the best performance with an average score of 55.07. This demonstrates that SAS's performance superiority holds across a range of data sizes, from small to large.

**SAS works well from small model size (e.g. 350M) to larger model size (e.g. 1.5B).** SAS consistently performs competitively. When the model size is 350M, SAS excels, topping most benchmarks (e.g. ARC-E, Hellaswag and Winograde), with an average score of 52.13. TPA also performs well with an average score of 50.51, but SAS shows broader dominance. When the model size becomes 760M or 1.5B, SAS still achieves better performance than other methods, with average score of 54.27 and 55.69. This demonstrates that SAS's performance benefits persist with larger model sizes.

## 5 Conclusion

In this work, we find that the attention head number and hidden size per head play crucial roles in the Transformer model's performance. Therefore, we propose SAS to simulate both the attention head

number and feature dimension per head to improve the performance. To control the parameter cost, we also propose the PEFA to aggregate attention outputs from each head. We conduct extensive experiments to validate our methods, including different lengths, different datasets, and different model sizes. Additionally, we scale up SAS and evaluate on downstream tasks, proving the effectiveness of our method. We believe that this paper provides an insight into using moderately sized models to mimic larger model attention score with enhanced expressiveness.

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

## A  Limitation

Compared to the baseline method, SAS may have higher computational costs due to the additional nonlinear mappings and increased attention heads and feature dimensions. Specifically, SAS employs a greater number of simulated attention heads, each with an increased hidden dimension, leading to a larger overall parameter count and more extensive matrix operations during both forward and backward passes.

## B  Broader Impacts

In this work, we introduce and develop a novel, more powerful approach for advancing language modeling capabilities. The positive broader impact of this research lies in its potential to find the way for the creation of even more effective models in the future, thereby contributing to progress in natural language processing and AI. However, alongside these benefits, it is also crucial to consider the potential negative broader impacts, emphasizing the importance of responsible usage, ethical considerations, and proactive measures to mitigate any unintended consequences associated with the deployment of such technology.

## C  Model Configuration

For the experiment on Arxiv and Books3 dataset, the 125M and 350M model configurations are as follows.

Table 2: The model configuration of different models follows the setting of TPA [100].

| Model | Hidden Size | Head | Hidden Size Per Head | Group Size |
|---|---|---|---|---|
| *125M parameters* | | | | |
| MHA | 768 | 12 | 64 | - |
| MQA | 768 | 23 | 64 | 23 |
| GQA | 768 | 22 | 64 | 11 |
| MLA | 768 | 13 | 64 | - |
| TPA | 768 | 36 | 64 | - |
| SAS | 768 | 36 | 96 | - |
| *350M parameters* | | | | |
| MHA | 1024 | 16 | 64 | - |
| MQA | 1024 | 31 | 64 | 31 |
| GQA | 1024 | 30 | 64 | 15 |
| MLA | 1024 | 23 | 64 | - |
| TPA | 1024 | 48 | 64 | - |
| SAS | 1024 | 48 | 96 | - |

For the experiment on the Fineweb-Edu dataset, the experiment setup is as follows: We follow the nanoGPT training configuration. In particular, we use the AdamW [44] optimizer with $(\beta_1, \beta_2) = (0.9, 0.95)$, a weight decay of $0.1$, and gradient clipping at $1.0$. We follow the same setting as nanoGPT that the learning rate is managed by a cosine annealing scheduler [43]. For the model setting, we mostly follow the setting of TPA [100].

## D  Experiment Statistical Significance

In this section, we have shown the mean and standard deviation of different methods, with three random seeds. According to the above table, we can find that our proposed method consistently achieves the best performance. All methods show the expected trend of decreasing perplexity with increased training steps. However, SAS maintains a consistent advantage throughout the training process, with the performance gap remaining relatively stable. And our method is significantly better than all other methods, with p value $< 0.05$. Therefore, we could have confidence to say that our SAS is significantly better than other proposed methods.

Table 3: The mean and standard variance of perplexity metrics of different methods on three random seeds, with training length 512 and the Arxiv dataset.

| Model | | 5K | 10K | 15K | 20K | 25K | 30K | 35K | 40K | 45K | 50K |
|---|---|---|---|---|---|---|---|---|---|---|---|
| MHA | Mean | 9.1802 | 8.0348 | 7.4585 | 7.0573 | 6.7825 | 6.4648 | 6.161 | 6.0311 | 5.956 | 5.8628 |
| | Std | 0.146 | 0.0593 | 0.0644 | 0.0532 | 0.0161 | 0.0629 | 0.1094 | 0.083 | 0.046 | 0.04200 |
| MQA | Mean | 9.1758 | 8.0427 | 7.4561 | 7.0629 | 6.7943 | 6.4823 | 6.182 | 6.0511 | 5.9797 | 5.8921] |
| | Std | 0.0802 | 0.0747 | 0.0452 | 0.0453 | 0.028 | 0.0655 | 0.0979 | 0.0844 | 0.0248 | 0.0212 |
| GQA | Mean | 9.2267 | 8.0921 | 7.5054 | 7.1177 | 6.845 | 6.534 | 6.2292 | 6.1036 | 6.0298 | 5.9385 |
| | Std | 0.1368 | 0.0428 | 0.0394 | 0.0538 | 0.0329 | 0.0776 | 0.0869 | 0.097 | 0.0244 | 0.0257 |
| MLA | Mean | 9.1931 | 8.1181 | 7.533 | 7.1373 | 6.8651 | 6.5456 | 6.2372 | 6.1044 | 6.0331 | 5.9416 |
| | Std | 0.1596 | 0.0888 | 0.0796 | 0.0325 | 0.0078 | 0.0516 | 0.1235 | 0.0727 | 0.0483 | 0.0382 |
| TPA | Mean | 9.059 | 7.9606 | 7.3845 | 7.0129 | 6.7562 | 6.4541 | 6.1567 | 6.0415 | 5.9695 | 5.8811 |
| | Std | 0.1232 | 0.0583 | 0.0409 | 0.06 | 0.0226 | 0.0722 | 0.1037 | 0.0922 | 0.0403 | 0.0363 |
| SAS | Mean | **8.6952** | **7.6776** | **7.1403** | **6.7776** | **6.5234** | **6.2342** | **5.9493** | **5.8368** | **5.7662** | **5.6821** |
| | Std | 0.1546 | 0.0286 | 0.0459 | 0.0623 | 0.0228 | 0.0703 | 0.1038 | 0.0854 | 0.0486 | 0.0425 |

# E  The SAS Performance with Varying Kernel Sizes

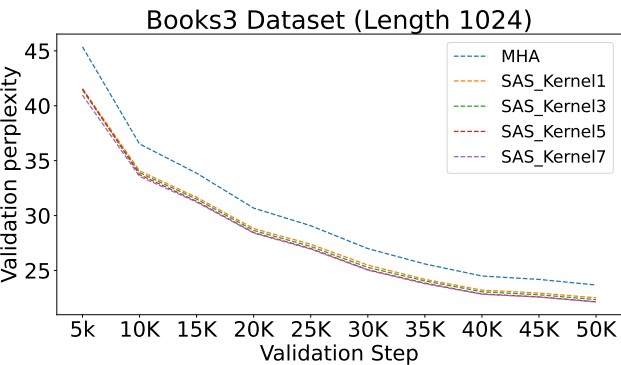

Figure 7: The performance of SAS with different kernels, with the Books3 dataset and training length 1024.

# F  The Training Cost

Table 4: The Time cost of different methods with training length 512 and batch size 1.

| Model | 125M | 350M | 2.7B | 6.7B | 10.6B |
|---|---|---|---|---|---|
| MHA | 36.20 | 72.87 | 117.82 | 186.58 | 217.01 |
| GQA | 38.87 | 71.66 | 133.76 | 205.69 | 239.16 |
| MQA | 39.37 | 72.01 | 132.62 | 202.61 | 238.85 |
| MLA | 51.68 | 104.95 | 180.03 | 254.43 | 637.78(Zero3 to avoid OOM;H200) |
| TPA | 44.19 | 86.56 | 151.71 | 235.85 | 258.41 |
| SAS | 68.07 | 138.53 | 215.13 | 350.70 | 402.15 |

# G  Large Mode Size 6.7B and 10.6B performance

Table 5: The perplexity of 6.7B and 10B on the Pile dataset.

| Model | 5K | 10K | 15K | 20K | 25K | 30K | 35K | 40K | 45K | 50K |
|---|---|---|---|---|---|---|---|---|---|---|
| 6.7B | | | | | | | | | | |
| MHA | 23.45 | 17.25 | 14.51 | 13.21 | 12.07 | 11.21 | 10.52 | 10.28 | 9.91 | 9.73 |
| GQA | 21.62 | 16.35 | 13.93 | 12.74 | 11.69 | 10.88 | 10.24 | 10.01 | 9.65 | 9.49 |
| MQA | 21.56 | 16.32 | 13.93 | 12.73 | 11.70 | 10.90 | 10.24 | 10.02 | 9.66 | 9.49 |
| MLA | 22.22 | 16.67 | 14.26 | 13.05 | 11.99 | 11.19 | 10.53 | 10.29 | 9.91 | 9.74 |
| SAS | 20.44 | 15.32 | 13.17 | 12.15 | 11.21 | 10.51 | 9.93 | 9.73 | 9.40 | 9.25 |
| 10.6B | | | | | | | | | | |
| MHA | 23.67 | 17.34 | 14.59 | 13.18 | 12.00 | 11.15 | 10.46 | 10.20 | 9.82 | 9.64 |
| GQA | 21.86 | 16.45 | 13.92 | 12.70 | 11.61 | 10.81 | 10.17 | 9.94 | 9.58 | 9.41 |
| MQA | 22.13 | 16.55 | 14.03 | 12.80 | 11.70 | 10.88 | 10.21 | 9.98 | 9.61 | 9.43 |
| MLA | 22.40 | 16.67 | 14.22 | 12.99 | 11.90 | 11.11 | 10.42 | 10.18 | 9.83 | 9.66 |
| SAS | 20.53 | 15.12 | 12.92 | 11.90 | 10.98 | 10.29 | 9.71 | 9.51 | 9.18 | 9.04 |

# H  Large Training Length performance

Table 6: The perplexity of length 16384 on the Pile dataset

| Model | 5K | 10K | 15K | 20K | 25K | 30K | 35K | 40K | 45K | 50K |
|---|---|---|---|---|---|---|---|---|---|---|
| MHA | 18.31 | 14.50 | 12.85 | 11.96 | 11.45 | 11.11 | 10.44 | 10.37 | 9.95 | 9.94 |
| GQA | 17.74 | 14.40 | 12.86 | 11.96 | 11.47 | 11.14 | 10.45 | 10.39 | 9.99 | 9.98 |
| MQA | 18.02 | 14.46 | 12.82 | 11.96 | 11.44 | 11.10 | 10.46 | 10.38 | 9.96 | 9.99 |
| MLA | 17.74 | 14.40 | 12.86 | 11.96 | 11.47 | 11.14 | 10.45 | 10.39 | 9.99 | 9.98 |
| TPA | 18.49 | 14.54 | 12.96 | 12.08 | 11.59 | 11.26 | 10.62 | 10.58 | 10.16 | 10.18 |
| SAS | 14.93 | 12.45 | 11.23 | 10.56 | 10.16 | 9.90 | 9.32 | 9.28 | 8.95 | 8.97 |

# I  Theoretical Analysis

The Softmax Attention with Fully Parameterized Bilinear Attention $O_i = \sum_{c=1}^{H} \left( \sum_{j=1}^{i} \phi(\frac{x_i W_c x_j}{\sqrt{D}}) x_j U_c \right)$ where: $W_c \in \mathbb{R}^{D \times D}$ = learnable bilinear transformation for head $c$, $\phi$ = the kernel function such as softmax, $U_c \in \mathbb{R}^{D \times D}$ = value projection matrix, $H$ = number of attention heads, $D$ = hidden dimension per head. Each head computes a weighted sum of values, with weights determined by $x_i W_c x_j^T$. For $m$-dimensional $x_i$ and $n$-dimensional $x_j$, one potential relation pattern is related to $x_i^m x_j^n$. Therefore, at most $D^2$ relations or interactions).

The bilinear term $x_i W_c x_j^T = \sum_{m=1}^{D} \sum_{n=1}^{D} x_i^{(m)} W_c^{(m,n)} x_j^{(n)}$ can be expressed as $(x_i \otimes x_j) \cdot \text{vec}(W_c)$, where $\text{vec}(W_c) \in \mathbb{R}^{D^2}$. With $H$ heads, the model can represent up to $D^2$ distinct relations [6]. We hope that the output of each head is a single relation but not mixed relations, so that we can utilize the relations independently (e.g., $\phi$ processes the relation independently). Assume $Rank(W_c) = 1$, and there is only one non-zero value in $W_c$. And there are $H = D^2 = 4$. $x_i = [a\ b]$ $x_j = [c\ d]$. $W_1 = [1\ 0]\ [0\ 0]$ $W_2 = [0\ 1]\ [0\ 0]$ $W_3 = [0\ 0]\ [1\ 0]$ $W_4 = [0\ 0]\ [0\ 1]$. The first head $W_1$ will gives the result of relation $ac$, the $W_2$, $W_3$, $W_4$ will give the result of $ad$, $bc$, $bd$. Therefore, this ($ac$, $ad$, $bc$, $bd$) suggests different attention patterns and relations. And they could be adjusted independently and not mixed (e.g., $\phi$ processes the relation independently), compared to only one head with $W_c = [1\ 1\ 1\ 1]$ that outputs the mix of $ac + ad + bc + bd$. With a large $D$, we can have more combinations. With a large $H$, we can capture more independent relations. If we increase $D$ (e.g., hidden size), we increase the maximum number of relations. If we increase the $H$, we increase the number of independent relations that are actually captured.

## J    SAS with Different Attention Types

Table 7: The perplexity of SAS with different methods, with training length 512 and Books3 dataset.

| Model | 5K | 10K | 15K | 20K | 25K | 30K | 35K | 40K | 45K | 50K |
|---|---|---|---|---|---|---|---|---|---|---|
| MHA | 60.0206 | 48.1325 | 42.9901 | 39.9132 | 36.6749 | 34.7711 | 32.1238 | 30.1071 | 30.019 | 29.8049 |
| MHA-SAS | **55.0498** | **45.1116** | **40.5295** | **37.6670** | **34.6701** | **32.9337** | **30.4943** | **28.6240** | **28.5052** | **28.3746** |
| MQA | 59.8441 | 48.4041 | 43.2021 | 40.128 | 36.9926 | 35.0613 | 32.4715 | 30.48 | 30.3328 | 30.1535 |
| MQA-SAS | **55.5273** | **45.7848** | **41.2885** | **38.4768** | **35.3813** | **33.6372** | **31.1418** | **29.2193** | **29.1556** | **29.0052** |
| GQA | 60.1109 | 48.2306 | 43.01 | 39.8758 | 36.5799 | 34.6976 | 32.0481 | 30.0991 | 29.9904 | 29.8059 |
| GQA-SAS | **55.1054** | **45.5153** | **41.0740** | **38.1912** | **35.2073** | **33.3761** | **30.9471** | **29.0752** | **28.9995** | **28.7853** |
| MLA | 60.4833 | 48.8735 | 43.6705 | 40.486 | 37.1407 | 35.2178 | 32.4767 | 30.4295 | 30.3255 | 30.1669 |
| MLA-SAS | **57.6595** | **46.8414** | **41.9994** | **38.9989** | **35.8767** | **34.0236** | **31.5449** | **29.5964** | **29.5283** | **29.3722** |
| TPA | 58.7126 | 47.9701 | 43.1154 | 40.1808 | 37.0918 | 35.3126 | 32.6225 | 30.6120 | 30.5273 | 30.3846 |
| TPA-SAS | **55.5189** | **45.2898** | **40.8298** | **37.9846** | **35.0361** | **33.2587** | **30.8783** | **28.9701** | **28.8586** | **28.7323** |

We present the results of SAS with varying attention types and model size of 125M, in Table 7. The MHA-SAS has triple attention heads of MHA with kernel size 5, while other SAS variants have double attention heads for their baseline with kernel size 1 as other variants have larger attention head numbers. We observe that SAS demonstrates superior performance relative to its corresponding standard attention types. Therefore, the SAS is compatible with most current attention methods to further enhance the performance.

## K    The Experiment Result on Large Model and Dataset

Table 8: The performance of different models with 1.5B model and 50B tokens. The performance of MHA, MQA, GQA, MLA and TPA comes from TPA paper [100]

| Method | ARC-E | ARC-C | BoolQ | HellaSw. | OBQA | PIQA | W.G. | MMLU | SciQ | Avg. |
|---|---|---|---|---|---|---|---|---|---|---|
| MHA | 64.81 | 35.41 | 61.90 | 54.32 | 37.20 | 72.74 | 55.80 | 25.44 | **82.80** | 54.49 |
| MQA | 64.10 | 36.01 | 62.26 | 54.38 | 39.00 | 72.58 | 56.43 | 23.70 | 81.90 | 54.48 |
| GQA | 63.68 | 35.92 | 60.46 | 54.17 | 38.40 | 73.56 | 56.27 | 24.77 | 81.70 | 54.33 |
| MLA | 64.14 | 35.92 | 60.12 | 53.60 | 39.20 | 72.25 | 55.17 | 24.71 | 81.60 | 54.08 |
| TPA | 66.71 | 36.52 | 61.38 | 54.03 | 40.40 | 72.52 | 56.83 | 24.49 | 82.20 | 55.01 |
| SAS | **66.84** | **37.88** | **61.41** | **56.05** | **41.60** | **72.52** | **57.85** | **26.11** | 82.60 | **55.85** |

## L    Implementation

In this section, we present the implementation of the proposed SAS module in `PyTorch` for research purposes, which is consistent with the intended use [49].

```python
import torch
import torch.nn as nn
import torch.nn.functional as F

class ResidualCNN(nn.Module):

    def __init__(
            self, input, output, kernel_size, padding
    ):
        super().__init__()

        self.ReLU = nn.ReLU()
        self.process = nn.Conv1d(input, output, kernel_size, 1,
            padding)

    def forward(self, hidden_states):
        output = self.ReLU(hidden_states)
        output = self.process(output)
        output = output + hidden_states
```

```python
            return output

class ResidualMLP(nn.Module):

    def __init__(
            self, input, output
    ):
        super().__init__()

        self.ReLU = nn.ReLU()
        self.process = nn.Linear(input, output)

    def forward(self, hidden_states):
        output = self.ReLU(hidden_states)
        output = self.process(output)
        output = output + hidden_states
        return output

class SAS(nn.Module):
  def __init__(self, ori_head, target_head, ori_feature, target_feature
      , kernel_size):
    """
    SAS attention bias module.

    Args:
      ori_head: the original head number
      target_head: the target head number
      ori_feature: the original feature size
      target_feature: the target feature size
      kernel_size: the kernel size
    """
    super(SAS, self).__init__()

    self.ori_head=ori_head
    self.target_head = target_head
    self.ori_feature=ori_feature
    self.target_feature = target_feature
    self.kernel_size = kernel_size
    padding = (kernel_size - 1) // 2
    self.sas_q = nn.Sequential(
        nn.Conv1d(self.ori_head, self.target_head, kernel_size, 1,
            padding),
        ResidualCNN(self.target_head, self.target_head, kernel_size,
            padding),
        nn.Linear(self.ori_feature, self.target_feature),
            ResidualMLP(self.target_feature, self.target_feature))
    self.sas_k = nn.Sequential(
        nn.Conv1d(self.ori_head, self.target_head, kernel_size, 1,
            padding),
        ResidualCNN(self.target_head, self.target_head, kernel_size,
            padding),
        nn.Linear(self.ori_feature, self.target_feature),
            ResidualMLP(self.target_feature, self.target_feature))
    self.sas_v = nn.Sequential(
        nn.Conv1d(self.ori_head, self.target_head, kernel_size, 1,
            padding),
        ResidualCNN(self.target_head, self.target_head, kernel_size,
            padding),
    )

    self.output_dense=nn.Linear(self.ori_head*self.ori_feature, self.
        ori_head*self.ori_feature)

  def forward(self, query, key, value):
    """
```

```
    Args:
      query: query embedding,
          shape [bsz, seq_len,num_heads, hidden_size_per_head]
      key: key embedding,
          shape [bsz, seq_len,num_heads, hidden_size_per_head]
      value: value embedding,
          shape [bsz, seq_len,num_heads, hidden_size_per_head]

    Returns:
      attention: attention output
          shape [bsz, seq_len,num_heads*hidden_size_per_head]
    """
    B, T, H,D = query.size()
    query = query.reshape(B * T, self.ori_head, self.ori_feature)
    key = key.reshape(B * T, self.ori_head, self.ori_feature)
    value = value.reshape(B * T, self.ori_head, self.ori_feature)

    #########Simulate Attention Score
    query = self.sas_q(query).reshape(B, T, self.target_head, self.
        target_feature)
    key = self.sas_k(key).reshape(B, T, self.target_head, self.
        target_feature)
    value = self.sas_v(value).reshape(B, T, self.target_head, -1)

    attention= F.scaled_dot_product_attention(query.transpose(1, 2),
        key.transpose(1, 2), value.transpose(1, 2), is_causal=True)

    ##########Parameter-Efficient Attention Aggregation
    attention = attention.transpose(1, 2).contiguous().view(B, T,
        self.target_head // self.ori_head, self.ori_head*self.
        ori_feature)
    attention = self.output_dense(attention)
    attention = attention.mean(dim=-2)

    return attention
```

