# OpenReview forum: "SAS: Simulated Attention Score"
_NeurIPS.cc/2025/Conference — NeurIPS 2025 poster_

### Official Review · Reviewer_DaRN · 2025-06-28

**Clarity:** 2
**Significance:** 2
**Originality:** 2
**Rating:** 3
**Confidence:** 3

**Summary:**

This paper proposes Simulated Attention Score, a technique that upscales the number of attention heads by applying a 2-layer residual 1-d convolution over the existing K, Q, and V tensors that increases the head dimension. It also uses a 2-layer residual MLP to increase the dimensionality of the K and Q tensors for the purposes of computing the attention weights.

The introduction motivates this idea of upscaling the number of attention heads through an additional feedforward layer with non-linearity.

The presentation is confusing. The authors in 3.3 begin with a detailed presentation of an MLP-based dimension upscaling for the heads. They then switch at line 162 to present a new convolution-based expansion of heads. I’m not sure, and the text gives little clue about how these two approaches relate, but I think (based on the code in Appendix G) the first MLP-based version is not used, and the second convolution-based one is used. Perhaps the authors can clarify this, and if it’s correct, can remove the initial section that’s not used.

As I understand the technique, the idea is as follows:
* Apply a couple of convolutional layers to the HxD block corresponding to the query tensor, producing an H’xD output with additional heads.
* Then apply a 2-layer MLP to this producing an H’xD’ ouptut with each new head having an upscaled dimension. This is the new query tensor
* Do the same to the key tensor
* Likewise for the value tensor, but without the upscaling of the feature dimension -- just increase the head dimension
* Compute attention based on these new tensors
* Mean-pool blocks of H’/H attention values to produce the final H-dimensional attention output
Otherwise, the transformer operates in the standard manner.

**Questions:**

It appears from Appendix F that the training time of SAS is significantly higher than all other methods, except in the case of the larger 2.7B model. This raises several questions:
Why is this true? Why is SAS slower to train for smaller model sizes? And what changes in the 2.7B model?
How do these numbers change for batch size >1 – setting batchsize 1 doesn’t seem consistent with best practices for pre-training.
What happens to the comparisons at equivalent wallclock training time? That is, if we increase the number of MHA heads until the training time on 350M is equivalent to SAS, how does the performance compare? Or, maybe a more general way to ask this: given a higher budget of clock training time, can you argue that SAS is the best way to use this additional budget, compared to other alternatives including adding heads or layers or width or feature dimensionality to previous techniques?

Some other questions:
Why did you adopt the mean pooling to aggregate the attention scores? Is upscaling the full H’ outputs back to the original dimensionality of the next transformer MLP too expensive? Was this decision the result of experiments showing this was a necessary approach?
Could you please clarify the number of weights in the convolution layers? Is it H_{in} * H_{out} for the first layer, and H_{out} * H_{out} for the second one?
Why upscale the head dimension before the feature dimension? Does the opposite approach not work as well?

**Ethical Concerns:**

["NO or VERY MINOR ethics concerns only"]

**Final Justification:**

I think the analysis is thorough and the authors did a great job through the rebuttal. I'd love to see this paper appear. However, I still have a couple of points that cause me to retain my original rating. SAS appears to introduce an increase in training overhead per step in order to improve the test metrics as a function of number of training steps. As to whether this is a win overall, the question remains open:

* For the benchmarks given by the authors, the reduction in perplexity seems to be equivalent to non-SAS with about 20K additional training steps -- in general, as SAS seems to add 2x-4x overheard per training step, this is generally not a favorable trade-off
* As the authors point out, if the final converged quality is better under SAS, the additional overhead may well be worth it. Unfortunately, I don't think any experiments demonstrate this, as they don't seem to have reached convergence.
* The authors give one example in which SAS performs better than non-SAS at equivalent wall-clock time: it is the 2.7B SAS model compared to the larger (10B) MHA model. This is a good example, but the larger models for this training setup don't seem to provide much benefit over the smaller ones. That is, one can always use a model that is too big in order to provide an example with very high wall-clock time and low quality, and this may be the situation.

**Limitations:**

I don't see any concerns on negative societal effects. The contriubtion represents a small change to the dynamics of the attention layer.

**Paper Formatting Concerns:**

In the main equations (2--12), there are a number of times the subscript h is used (as in Q_h). I'm not sure why this subscript is unchanged for the settings before and after the number of heads changes. This may be the authors' intention, though. For example, equation 5 vs 7 both use Q_h for Conv_1 and Conv_2 respectively, even though the dimensionalities of the operators are different.

**Quality:**

2

**Strengths And Weaknesses:**

Strengths:
* The approach is straightforward and easy to implement
* The performance analysis shows significant improvements in perplexity, though I have a number of questions below
* There is some evidence of increased performance on downstream tasks
Weaknesses:
* There are a number of decisions that are not fully motivated
* The wallclock time for training is significantly longer for SAS than the competitors. The baselines are not scaled appropriately.
* There is little analysis of total number of parameters, though this is likely to be favorable for SAS
* The originality of the approach is not high. The decision about how to upscale the heads and hidden attention dimensions is interesting (convolution in 1 case, MLP in the other), but not studied or fully motivated
* Clarity of presentation is adequate but could be improved
* Significance of findings misses a number of questions that are necessary to substantiate the improvements

---

> ### Author Rebuttal · Authors · 2025-07-31
>
> Dear Reviewer DaRN,
>
> Thank you very much for your comment. We will address your concerns below.
>
> **Q1: The discussion of the time cost**
>
> A: Thank you very much for the notice. We recheck and retest the time cost, and find that some 2.7B results of baseline may be recorded with batch size 4 (the GPT-NeoX default training batch size), or the GPU cluster is also occupied by other programs during the speed test. **For the updated test, we use a local standalone machine that has 80GB-HBM3 H100 GPUs**. The following are the updated results.**
>
> *The time cost with different model sizes and micro batch size 1, with H100*
> | Method | 125M  | 350M   | 2.7B    | 6.7B|10.6B|
> |--------|--------|---------|----------|----------|----------|
> | MHA    | 36.20  | 72.87   | 117.82   |186.58|217.01|
> | GQA    | 38.87  | 71.66   | 133.76   |205.69| 239.16|
> | MQA    | 39.37  | 72.01   | 132.62   |202.61| 238.85|
> | MLA    | 51.68  | 104.95  | 180.03   |254.43|637.78(Zero3 to avoid OOM;H200)
> | TPA    | 44.19  | 86.56   | 151.71   |235.85|258.41|
> | SAS    | 68.07  | 138.53  | 215.13   |350.70|402.15|
>
>
> *The time cost with different model sizes and micro batch size 4*
> | Method | 125M  | 350M   | 2.7B    |
> |--------|--------|---------|----------|
> | MHA    | 38.01  | 77.34   | 186.03   |
> | GQA    | 45.26  | 78.59   | 245.86   |
> | MQA    | 44.07  | 76.23   | 243.67   |
> | MLA    | 53.40  | 108.65  | 258.06   |
> | TPA    | 56.37  | 116.04  | 318.31   |
> | SAS    | 89.70  | 200.47  | 531.17   |
>
> * The computional cost cost of MHA:$O(12Th^2d^2)+O(T^2hd)$, while $h$ is the attention head number, $d$ is the hidden size per head and $T$ is the sequence length.
> * The computional cost of SAS: $O(11T h^2 d^2+Th'^2d'^2)+O(T^2h'd')$, while $h'$ is the simulated attention head number, $di$ is the simualted hidden size per head and $T$ is the sequence length.
> * Usually, the $h'/h$ is between 1 to 3, and the $d'/d$ is between 1 to 1.5 (less than 1.2 for larger models to control cost)
>
>
>
>
>
>
>
> **The discussion of the next step of the model architecture**
> * **The computational cost becomes cheaper and cheaper**. With the development of GPUs, the computational cost becomes cheaper and cheaper. This is the basic reason that Large Language Models could be true.
> * **The high-quality data is limited**. The current Large Language Model almost uses all the web data.
> * **Therefore, we have to develop more powerful models with higher expressiveness**. Compared to linear models (e.g. LSTM, RNN), the Transformer has higher cost but performance. And now, most STOA models usually use the Transformer. There are also recent works that propose the model with cost $T^3$ (e.g., DeltaFormer, Simplex Attention), where the $T$ is the sequence, while the Transformer has cost $T^2$. **More powerful with higher cost may be the next-generation architecture.**
>
> **Q2: Or, maybe a more general way to ask this: given a higher budget of clock training time, can you argue that SAS is the best way to use this additional budget, compared to other alternatives including adding heads or layers or width or feature dimensionality to previous techniques?**
>
>
>
> A2: **We can be sure that using the same parameter number budget, the SAS could achieve the BEST Performance, compared to previous techniques. With the comparable budget of clock training time, the SAS has the potential to have better performance**
> * **With the same parameter number budget, the SAS could have the best performance**. This has already been discussed in the paper experiment part.
> * **With the same/comparable time, the SAS has the potential to have better performance.**
>
> *The following is the perplexity performance with 6.7B and 10.6B.*
> | Model                     |5K|10K|15K|20K|25K|30K|35K| 40K|45K|50K|
> |-----|-----|-----|-----|-----|-----|-----|-----|-----|-----|-----|
> |6.7B|
> | MHA                 | 23.45 | 17.25 | 14.51 | 13.21 | 12.07 | 11.21 | 10.52 | 10.28 | 9.91 | 9.73|
> | MQA |21.56 | 16.32 | 13.93 | 12.73 | 11.70 | 10.90 | 10.24 | 10.02 | 9.66 | 9.49|
> |GQA|21.62 | 16.35 | 13.93 | 12.74 | 11.69 | 10.88 | 10.24 | 10.01 | 9.65 | 9.49|
> |MLA|22.22 | 16.67 | 14.26 | 13.05 | 11.99 | 11.19 | 10.53 | 10.29 | 9.91 | 9.74|
> | SAS         | **20.44** | **15.32** | **13.17** | **12.15** | **11.21** | **10.51** | **9.93** | **9.73** | **9.40** | **9.25**|
> |10.6B|
> | MHA                 | 23.67 | 17.34 | 14.59 | 13.18 | 12.00 | 11.15 | 10.46 | 10.20 | 9.82 | 9.64|
> | MQA |22.13 | 16.55 | 14.03 | 12.80 | 11.70 | 10.88 | 10.21 | 9.98 | 9.61 | 9.43|
> | GQA            | 21.86 | 16.45 | 13.92 | 12.70 | 11.61 | 10.81 | 10.17 | 9.94 | 9.58 | 9.41|
> |MLA|22.40 | 16.67 | 14.22 | 12.99 | 11.90 | 11.11 | 10.42 | 10.18 | 9.83 | 9.66|
> | SAS         | **20.53** | **15.12** | **12.92** | **11.90** | **10.98** | **10.29** | **9.71** | **9.51** | **9.18** | **9.04**|
>
>
> * With the same parameter cost, the SAS is definitely better than other methods. We have proved this in the paper.
> * With the comparable clocking time, the SAS has the potential to have better performance. For example, the SAS with 6.7B is significantly better than MHA, GQA, MQA, MLA with 10.6B.
>
> **Adding Head**
> * We have discussed the effect of attention head in Figure 1. With MHA and the same parameter cost and hidden size, the SAS is always better than MHA, whatever the attention head number is 1 or 96.
>
> **Adding Layer or Width or Feature Dimensionality**.
> * Such an operation will lead to a higher model parameter cost. **Our setting is using the same parameter number to achieve better performance**
>
>
> The discussion of previous works
> * **Mixture-of-Experts (MoE) could achieve better performance than da ense model with the same activated parameters, while the MoE needs more parameter cost.**
> * **Transformer usually is better than a linear model with the same parameter cost, while Transformer needs a quadratic cost of sequence length and more time cost.**
> * **Similarly, this work targets to improve the performance with the same/comparable parameter number**.
>
> **Q3: Why did you adopt the mean pooling to aggregate the attention scores? Is upscaling the full H’ outputs back to the original dimensionality of the next transformer MLP too expensive?**
>
> A3: **The Attention output feature dimension should still be the hidden size (e.g., 768 for 125M). Also, we have to control the number so that we can use mean pooling. Ifwe directly map the H' back to the original dimension, the parameter may increase by 25%. Our setting is using the same parameter number to achieve better performance so that we do not directly map the full H' back.
>
> Why average pooling (use 125M as an example):
> * Originally, the attention output will have shape [12 64] so that we could reshape it to [768] and then multiply a matrix with shape [768 768].
> * For SAS, the attention output will have [36 64]. Therefore, we have to first reshape it to [3 12 64] and then average along the first dimension to have [1 12 64]. Then, we reshape it to have [768] and multiply with the matrix [768 768].
>
> **Let us calculate the parameter cost for upscaling the full H outputs back**
> * Assume we simulate triple attention heads. Therefore, the additional parameter cost is $3D^2$.
> * The FFN takes $8D^2$ parameter cost.
> * The original Attention takes $4D^2$ cost.
> * Therefore, **upscaling the full H outputs back will take an addtional 25% parameter cost.**
> * **Our experiment setting uses the same parameter cost to improve the performance. Therefore, we cannot afford such an expensive additional parameter cost.**
>
>
> **Q4: Could you please clarify the number of weights in the convolution layers? Is it H_{in} * H_{out} for the first layer, and H_{out} * H_{out} for the second one?**
>
> A4: Yes, the understanding is correct. Finally, there are $H_{out}$ simulated attention heads.
>
> **Q5:Why upscale the head dimension before the feature dimension? Does the opposite approach not work as well?**
>
> A: Thank you very much for the suggestion. We first realize that more attention is necessary, and then extend it to feature expansion. And this is why we first process the attention head, and then the feature expansion. The following is the result of processing the feature dimension first and then the attention head.
>
> *The perplexity result on the Books dataset with length 512 and H100*
>
> | Model                     | 5K|10K|15K|20K|25K|30K|35K| 40K|45K|50K|
> |-----|-----|-----|-----|-----|-----|-----|-----|-----|-----|-----|
> | MHA                 | 59.23 | 46.31 | 44.68 | 39.33 | 36.57 | 35.44 | 33.19 | 30.35 | 30.29 | 28.68 |
> | SAS_FeatureFirst            | **53.98** | **43.44** | **42.35** | **37.27** | 34.86 | 33.94 | 31.80 | 29.15 | **29.06** | 27.63 |
> | SAS         | 54.34 | 43.56 | 42.36 | 37.31 | **34.73** | **33.84** | **31.70** | **29.13** | 29.07 | **27.52** |
>
> According to the above result, the SAS_FeatureFirst   achieves better performance at the early stage, while the SAS finally achieves better performance. This may be caused by that the SAS (head first) utilizes more computation cost.
> *  SAS_FeatureFirst:12x(64x96+96x96)+96x(12x36+36x36)=350208;
> * SAS: 64x(12x36+36x36)+36x(64x96x96x96)=663552.
>
> **Q6: or example, equation 5 vs 7 both use Q_h for Conv_1 and Conv_2 respectively, even though the dimensionalities of the operators are different.**
>
> A6: The $Q_h$ means the operation is for Query Head. The Conv1 is the first operation (e.g., mapping head from 12 to 36), and the Conv2 is the second operation (e.g., from head number 36 to 36). The $Q_h^{Conv1}$ is the first convolution operation for Query Head, so the $Q_h^{Conv2}$ is the second convolution operation for Query Head.
>
> **If the response addresses the concerns, could you consider improving the score?**

---

> > ### Comment · Reviewer_DaRN · 2025-08-08
> > **Response to authors**
> >
> > Thanks to the authors for their analysis, it is well done. Many of my issues are addressed -- thank you for the nice comparison of FeatureFirst. I still have a few questions:
> >
> > 1. If I read the tables above correctly, the performance is measured at training length up to 50K. However, the models do not appear to be fully converged at that point. Performance of MHA on Books, for instance, shows a significant improvement from 45K to 50K, and on Pile the same appears to be true. The argument for SAS performing better at the same number of parameters seems to be true at the same point in training, but this isn't a good comparison as the number of FLOPS is quite different between the two methods at the same point in training. Performance at fully converged state is important -- could you clarify the results to support the fully converged quality numbers?
> > 1. Thank you the discussion of performance at equal wallclock time. Your point regarding SAS at 2.7B compared to MHA at 10B is good. However, my concern on this argument is that given the training set, architecture, etc, the gains for increased model scale seem to be already saturating before 10B, so it seems possible that MHA at 10B is not optimal from a scaling law perspective -- perhaps the additional compute to support that model size is not well utilized. Can you make the same argument for other datasets where there is a clear advantage to increased model size, or for MHA versus SAS for smaller model sizes (ie, MHA 2.7B versus smaller SAS versions with the same wallclock time)?
> >
> > Thank you very much for your inputs

---

> ### Author Response · Authors · 2025-08-01
> **The performance of SAS 2.7B**
>
> Dear Reviewer DaRN,
>
> We further conducted an experiment of SAS with 2.7B, which has a comparable time cost to baseline MHA 10.6B.
>
> *The result on the Pile dataset with length 512 and H100*
>
>
>
> | Model                     | 5K|10K|15K|20K|25K|30K|35K| 40K|45K|50K| Time Cost|
> |-----|-----|-----|-----|-----|-----|-----|-----|-----|-----|-----|-----|
> | MHA 10.6B                 | 23.67 | 17.34 | 14.59 | 13.18 | 12.00 | 11.15 | 10.46 | 10.20 | **9.82** | **9.64**| 217.01 |
> | SAS 2.7B            | **20.67** | **15.82** | **13.66** | **12.66** | **11.72** | **10.99** | **10.38** | **10.19** | 9.85 | 9.70| 215.13|
> | SAS 10.6B        | 20.53 | 15.12 | 12.92 | 11.90 | 10.98 | 10.29 | 9.71 | 9.51 | 9.18 | 9.04| 402.15|
>
> * **In this work, we aim to achieve better performance with the same/comparable parameter cost. Therefore, we have confidence to claim that the SAS is definitely better than all other methods with the same parameter cost**
> * **With the same time cost, the SAS has the potential to achieve comparable performance. For example, SAS 2.7 achieves comparable performance with MHA 10.6B.**

---

> > ### Author Response · Authors · 2025-08-04
> > **Kind Reminder of Discussion Period**
> >
> > Dear Reviewer DaRN,
> >
> > Hope this finds you well.
> >
> > We have addressed the concerns in the above response, including the time cost, performance with the comparable time cost (2.7B SAS has comparable performance with 10.6B MHA), the discussion of mapping H', process feature dimension first and so on. As the discussion period will be closed soon, if possible, could you consider improving the score?

---

> ### Author Response · Authors · 2025-08-06
> **Kind Reminder of Discussion Period Ends**
>
> Dear Reviewer DaRN,
>
> We have addressed the concerns in the above response, including the time cost, performance with the comparable time cost (2.7B SAS has comparable performance with 10.6B MHA), the discussion of mapping H', process feature dimension first and so on. If there are any questions, please let us know before the discussion period ends (August 8 AoE). After the detailed explanation, if possible, could you consider improving the score?
>
> Again, thank you for your attention, and wish you a good day.

---

> > ### Author Response · Authors · 2025-08-08
> > **Discussion Will End In One Day**
> >
> > Dear Reviewer DaRN,
> >
> > Thank you for your attention and precious time. As the discussion will end in one day, could we know whether there are remaining concerns? If the above response addresses the concerns, could you consider improving the score?

---

> ### Author Response · Authors · 2025-08-08
>
> Dear Reviewer DaRN,
>
> Thank you for the message. We will answer your question below.
>
> **Q1: Performance of MHA on Books, for instance, shows a significant improvement from 45K to 50K. could you clarify the results to support the fully converged quality numbers?**
>
> A1: **We provide the large-scale training results with larger training steps to support that the SAS is still better than MHA. Also, we have to point out that as long as we further add training data/increase training steps (even with trillion tokens), the loss will gradually decrease (e.g., loss drops from 1.4 to 1.3 with training tokens increasing from 12T to 15T), which is supported by recent work Kimi K2 Figure 3 [1].**
>
> *The loss on FinWeb-Edu with training step 100K,  50K tokens, and cosine learning rate scheduler*
> | Model                     |10K|20K|30K|40K|50K|60K|70K| 80K|90K|100K|
> |-----|-----|-----|-----|-----|-----|-----|-----|-----|-----|-----|
> |350M 50B training tokens|
> |MHA| 3.64| 3.34 |  3.20 | 3.11 | 3.05 | 3.00 | 2.95| 2.90| 2.86|2.83|
> |SAS|**3.52** |**3.23**|  **3.10**| **3.02**| **2.97**| **2.91**| **2.87**| **2.83**| **2.79**|**2.76**|
>
> At the end of the training, the loss reduction is not significant compared to previous steps. For example, from 80K to 90K, the loss of MHA reduces by 0.04, while from 90K to 100K, the loss reduces only by 0.02. Also, even if we double the training steps, the SAS is still better than MHA. **We can be sure that with the same training steps, the SAS is always better than MHA. We have to point out that as long as adding training data/training steps, the loss will gradually decrease  (e.g., loss drops from 1.4 to 1.3 with training tokens increasing from 12T to 15T), as supported by Kimi K2.**
>
> **As long as the training length, training tokens and training steps, and others are the same, with the same parameter cost, we have confidence to say that SAS is better than MHA**.
>
>
> **Q2: Can you make the same argument for other datasets where there is a clear advantage to increased model size?**
>
> A2: **The main target of this work is to increase performance with the same Parameter Cost**.
>
> * **We have confidence to say that with the same parameter cost, we can be sure that SAS is better than MHA.**
> * With the same wall-clocking time, SAS has the potential to achieve comparable performance than MHA, and this depends on the model size, datasets, and so on.
>
> **The following question of Reviewer DaRN may be: if we only promise that SAS achieves better performance than SAS with the same parameter cost but not wall-clocking time, then why should we believe SAS is a good work?**
> * **Computational Cost becomes cheaper**. In the past several years, the GPUs have become more and more powerful, and we can train larger models at a larger cost.
> * **As we know, the Transformer is only able to solve the $NC^0$ problem [2]. Therefore, we need a more powerful model to further solve problems that cannot be solved by Transformer, even if the computational cost is higher**. Recently, there have also been other researchers agreed, so that they have developed models with $O(T^3)$ time cost, such as Deltaformer [2], Simplex Attention [3], and so on.
> * **Large Cost may be the future**.
>     * We increase more costs on the parameter so that we have the Mixture-of-Experts.
>    * We increase more costs on the model, so we have a Transformer from LSTM.
>    * We scale the model up with more cost, including data, model size, and so on, so we have a large language model.
>
> **We hope that the above insight could address the concerns. If there are any other questions, please let us know.**
>
> Reference:
>
> [1] Team, K., Bai, Y., Bao, Y., Chen, G., Chen, J., Chen, N., ... & Zhang, H. (2025). Kimi K2: Open Agentic Intelligence. arXiv preprint arXiv:2507.20534.
>
> [2] Zhong, S., Xu, M., Ao, T., & Shi, G. (2025). Understanding Transformer from the Perspective of Associative Memory. arXiv preprint arXiv:2505.19488.
>
> [3] Roy, A., Chou, T., Duvvuri, S. S., Chen, S., Yu, J., Wang, X., ... & Anil, R. (2025). Fast and Simplex: 2-Simplicial Attention in Triton. arXiv preprint arXiv:2507.02754.

---

> > ### Author Response · Authors · 2025-08-08
> >
> > Dear Reviewer DaRN,
> >
> > Thank you for the engagement in the discussion. Please let us know if there is anything that you would like to discuss, including but not limited to this work, but also our insight into future model architecture, the understanding of attention, and so on.
> >
> > Again, thank you for the attention, and wish you a good day.

---

> > ### Comment · Reviewer_DaRN · 2025-08-08
> > **Response to authors**
> >
> > Thank you, this is very helpful additional data. One way I could view this data on longer training runs is to say that, roughly, SAS with T training steps outperforms MHA with T training steps, but is similar to MHA with T+20k training steps. Given that the cost per training step for SAS seems to be 2-4x higher than MHA (depending on the batch size, and the batch size 4, which seems to show a larger gap based on your earlier tables, seems like a more standard training size), doesn't this suggest that rather than using SAS I should simply train MHA for longer? Thank you for your thoughts on this question

---

> > > ### Author Response · Authors · 2025-08-08
> > >
> > > Dear Reviewer DaRN,
> > >
> > > Thank you for the response. We will answer your question below.
> > >
> > > **Q1: doesn't this suggest that rather than using SAS I should simply train MHA for longer? Thank you for your thoughts on this question**
> > >
> > > A1: **This is a good question. But one thing is important: We may have to notice that the training data is not infinite.**
> > >
> > > **If the training data is infinite, then with the same wall-clocking time, we could train MHA longer for potentially better performance**
> > >
> > > **However, our training data is NOT Infinite.** Even with a trillion-level dataset, the current GPU cluster could finish the training.  **Then, a question is: after MHA uses all available training data, what could we do?** Could we further improve the performance? The answer is NO. **On the other hand, if we use SAS, we can further improve the performance. The current LLM already faces the problem of using all available data, which is why we are trying to synthesize data**.
> > >
> > > **To summarize**:
> > > * **If the data is infinite, then we can simply train MHA longer to potentially have better performance**
> > > * **However, practically, the dataset is not infinite. And LLM already faces the problems of using all the potential data. During this situation, we can use SAS to further improve the performance.**
> > >
> > > **If there are any questions, please let us know**.

---

> > > > ### Comment · Reviewer_DaRN · 2025-08-08
> > > > **response to authors**
> > > >
> > > > Thank you for your additional thoughts on this. I think there is absolutely an argument that SAS may be able to outperform other forms of attention when trained to convergence on limited data. It is also possible that, as the training moves to convergence, and gains become harder to attain, there comes a time when MHA will require significantly more training time to reach the same level of performance as SAS. I think both of these are good arguments for SAS, but I don't think they have been convincingly demonstrated by the experiments yet. The arguments that I think have been established are:
> > > > 1. SAS at equivalent training steps outperforms MHA, but the wall clock time is longer. For the datasets we have seen, I don't think there is a clear demonstration that for the same architecture and the same training time, SAS generally results in a model with better quality (perplexity or downstream) -- though it might be possible to produce this table based on data you already have: basically show for the same model size the test perplexity of MHA vs SAS at the same training time, even if the number of training steps is different. Would it be possible to produce this table for a few datasets and model sizes, as I think this is at the heart of the question?
> > > > 2. There are examples where SAS performs better than MHA for equivalent training time (2.7B vs 10B, for instance), but it is not clear whether the additional cycles that MHA uses in the 10B model are well spent, or whether the larger model is not helpful for that dataset.

---

> ### Author Response · Authors · 2025-08-08
>
> Dear Reviewer DaRN,
>
> Thank you very much for the message. We will answer your question.
>
> **Before the explanation, we would like to reclaim our argument:**
> * **With the same parameter cost, the SAS is definitely better than MHA, as long as others are the same, including training step, batch size, learning rate, and so on**.
> * With the same time cost, the SAS **only has the potential (but not promising)** to have comparable performance, as **MHA with the same time cost either has more training data or a larger model size.**
>
> **Q1: SAS at equivalent training steps outperforms MHA, but the wall clock time is longer. demonstration that for the same architecture and the same training time, even with smaller training step**
>
> A1: **Our main claim is: SAS at equivalent training steps outperforms MHA. But we never promise that SAS outperforms MHA with the same blocking time, but SAS only has the potential to have comparable performance with MHA with a comparable time cost.**
>
> We have to notice that, if we directly use previous results by reducing the training step, we also **change the other hyperparameters,  such as training data size. For example, when MHA is trained with 50K steps with 10.6B, if with the same time cost, the SAS with 10.6B is trained with 25K steps, which only.use half dataset. Therefore, forcing such comparison (same model size and time cost) is actually unfair, as too many other hyperparameters are changed, including learning rate, training dataset size and so on. On the other hand, compare MHA 10.6B and SAS 2.7B for same time cost is relatively acceptable , as the dataset size is not.changed**.
>
>
>
>
> *The training loss  on Pile with 50K training steps, 10.6B model size, with the each column result are reported with the same time cost*
> | Model     | MHA at 5K | MHA at 10K | MHA at 15K | MHA at 20K | MHA at 25K | MHA at 30K | MHA at 35K | MHA at 40K | MHA at 45K | MHA at 50K | SAS at 50K |
> |-----------|----------|-----------|-----------|-----------|-----------|-----------|-----------|-----------|-----------|-----------|-----------|
> | MHA 10.6B | 3.29     | 2.99      | 2.85      | 2.72      | 2.60      | 2.57      | 2.47      | 2.45      | 2.42      | **2.38**  | 2.38      |
> | SAS 2.7B  | **3.15** (5K Steps) | **2.89** (10K Steps) | **2.79** (15K Steps) | **2.68** (20K Steps) | **2.58** (25K Steps) | **2.56** (30K Steps) | **2.46** (35K Steps) | **2.44** (40K Steps) | **2.42** (45K Steps) | 2.39 (50K Steps) | 2.39 (50K Steps)-         |
> | SAS 10.6B | 3.64 (2.5K Steps) | 3.15 (5K Steps) | 2.98 (7.5K Steps) | 2.85 (10K Steps) | 2.75 (12.5K Steps) | 2.74 (15K Steps) | 2.67 (17.5K Steps) | 2.62 (20K Steps) | 2.57 (22.5K Steps) | 2.52 (25K Steps) | **2.32** (50K Steps) |
>
>
>
>
>
>
> **Q2: whether the larger model is not helpful for that dataset**
>
> A2: **SAS still significantly improves the performance with the increase of model size**
>
>
>
> *The perplexity on Pile*
> | Model                     |5K|10K|15K|20K|25K|30K|35K| 40K|45K|50K|
> |-----|-----|-----|-----|-----|-----|-----|-----|-----|-----|-----|
> |MHA|
> |6.7B| 23.45 | 17.25 | 14.51 | 13.21 | 12.07 | 11.21 | 10.52 | 10.28 | 9.91 | 9.73|
> |10.6B|23.67 | 17.34 | 14.59 | 13.18 | 12.00 | 11.15 | 10.46 | 10.20 | 9.82 | 9.64|
> |ppl change|-0.22 | -0.09 | -0.08 | 0.03 | 0.07 | 0.06 | 0.06 | 0.08 | 0.09 | 0.09|
> |SAS|
> | 6.7B         | 20.44 | 15.32 | 13.17 | 12.15 | 11.21 | 10.51 | 9.93 | 9.73 | 9.40 | 9.25|
> |10.6B|20.53 | 15.12 | 12.92 | 11.90 | 10.98 | 10.29 | 9.71 | 9.51 | 9.18 | 9.04|
> | ppl change                 | -0.09 | 0.20 | 0.25 | 0.25 | 0.23 | 0.22 | 0.22 | 0.22 | 0.22 | 0.21|
>
> **The model scale-up follows the suggestion of the GPT-3 paper.**
>
> | Model   | Layer Number|Hidden Size | Head Number | Hidden Size Per head|
> |---------|----------|-----------|-----------|-----------|
> |2.7B|
> | MHA     | 32|2560 | 32|80|
> | SAS     | 32|2560|96|96|
> |6.7B|
> | MHA     | 32|4096 | 32|128|
> | SAS     | 32|4096|96|144|
> |10.6B|
> | MHA     | 32 | 5120|40|128|
> | SAS     | 32|5120|120|144|

---

> ### Author Response · Authors · 2025-08-08
>
> Dear Reviewer DaRN,
>
> **Our Core Claim is: SAS at equivalent training steps outperforms MHA, with the same parameter cost. All experiments are designed for such a claim.**
> * This is readily proved in the paper experiment.
>
> **However, we never claim that SAS will definitely outperform MHA with the same training time, but SAS only has the potential to achieve comparable performance with MHA with the same time cost, depending on the dataset, model size, and so on.**
>    * 2.7B SAS achieves comparable performance with 6.7B MHA and 10.6B MHA.
>    * However, this is not always promised, depending on the dataset, model size, and so on.
>
> **Therefore, we both agree: 1) SAS at equivalent training steps outperforms MHA, with the same parameter cost; 2) SAS has the potential to achieve comparable performance with MHA with the same time cost (2.7B SAS and 10.6B MHA on Pile dataset), depending on model size, dataset, and so on.**
>
> **If there are any questions, please let us know**

---

> > ### Comment · Reviewer_DaRN · 2025-08-08
> > **response to authors**
> >
> > Thank you again for these additional comments, your discussion here makes sense to me

---

> > > ### Author Response · Authors · 2025-08-08
> > >
> > > Dear Reviewer DaRN,
> > >
> > > Thank you very much the engagement for the discussion. And thank you for the confirmation for the discussion. If the concerns are addressed, sincerely hope that the score could be improved.
> > >
> > > Again, thank you very much for the support, and wish you a good day.

---

> > > > ### Author Response · Authors · 2025-08-08
> > > >
> > > > Dear Reviewer DaRN,
> > > >
> > > > After the above discussion,.is there any concern left? If all concerns are addressed, could you consider improving the score?
> > > >
> > > > Again, thank you very much for the support, and wish you a good day.

---

> > > > > ### Comment · Reviewer_DaRN · 2025-08-09
> > > > >
> > > > > Thank you, I have a better understanding of the contributions. I will leave my score unchanged for now as we move into internal discussion.

---

> ### Author Response · Authors · 2025-08-09
>
> Dear Reviewer DaRN,
>
> We thank Reviewer  DaRN   for acknowledging our clarifications on the contributions. We are open to further modifications if any.

---

### Official Review · Reviewer_wQTE · 2025-06-30

**Clarity:** 2
**Significance:** 2
**Originality:** 3
**Rating:** 4
**Confidence:** 3

**Summary:**

In this paper, the authors introduce Simulated Attention Score to improve Transformer performance by simulating a larger number of attention heads and hidden feature dimensions without significantly increasing the model's parameter count.
SAS achieves this through non-linear mappings that project low-dimensional head and feature representations into higher-dimensional spaces.
The authors also propose Parameter-Efficient Attention Aggregation (PEAA) to manage parameter costs.
Experiments demonstrate that the proposed SAS outperforms existing attention mechanisms, achieving lower perplexity and faster convergence.

**Questions:**

See weaknesses

**Ethical Concerns:**

["NO or VERY MINOR ethics concerns only"]

**Final Justification:**

The authors provided very detailed supplementary experiments that effectively addressed all my concerns. Therefore, I have decided to raise my score to borderline accept to reflect this.

**Limitations:**

Yes

**Quality:**

2

**Strengths And Weaknesses:**

# Strengths

1. The main idea of using more attention heads and larger dimensions is simple and reasonable.

2. The experimental results show that the proposed SAS achieves promising improvements.


# Weaknesses

1. The writing of this paper is not clear and confuses me. For example:
    - In Section 3.1, the discussion of using only one attention head (when the base model has 12) or 24 heads (when the original is 12) is confusing and requires clarification.
    - In Sec. 4, we do not know the configuration of the larger models (350M and 2.7B). Although we know the configuration of 125M models, the authors do not explain the relationship between the configuration data and Equations 1-12. I don’t fully understand the dimension sizes for each part. A diagram might help clarify this.

2. SAS introduces additional layers and residual connections, yet there is no discussion of its impact on inference speed, particularly for long-context scenarios. This is a critical consideration for practical deployment.

3. The training lengths of 512 and 1024 are too short. It remains unclear whether SAS maintains its advantages in long-context settings (e.g., >=16K tokens).

4. A thorough double-checking of all reported metrics is necessary to ensure accuracy. For example, in Table 1, the 350M SAS model does not consistently outperform baselines (e.g., in SocialIQA and Winogrande), yet the authors highlight it in bold as the best performer.

---

> ### Author Rebuttal · Authors · 2025-07-31
>
> Dear Reviewer wQTE,
>
> Thank you very much for your comment. We will address your concerns below.
>
> **Q1: In Section 3.1, the discussion of using only one attention head (when the base model has 12) or 24 heads (when the original is 12) is confusing and requires clarification.**
> A1: According to Figure 1, we can find that with the attention head number gradually increasing and the hidden size per head gradually decreasing, the ppl first decrease and then increase. Then, we assume that if the hidden size is large enough, the performance may gradually increase as the attention head number increases.
> How can we make the attention head and hidden size per head large? We then propose to use non-linear mapping to make them larger, which is the simulated attention score.
>
> **Q2: We do not know the configuration of the larger models (350M and 2.7B). Relationship between the configuration data and Equations 1-12.**
>
> A2: A:  We further provide the configuration in the following.
> | **Model** | Hidden Size | Head | Hidden Size Per Head | Group Size |
> |-----------|-------------|------|----------------------|------------|
> | *125M parameters* |  |  |  |  |
> | MHA       | 768         | 12   | 64                   | -          |
> | MQA       | 768         | 23   | 64                   | 23         |
> | GQA       | 768         | 22   | 64                   | 11         |
> | MLA       | 768         | 13   | 64                   | -          |
> | TPA       | 768         | 36   | 64                   | -          |
> | SAS       | 768         | 36   | 96                   | -          |
> | *350M parameters* |  |  |  |  |
> | MHA       | 1024         | 16   | 64                   | -          |
> | MQA       | 1024        | 31   | 64                   | 31         |
> | GQA       | 1024         | 30   | 64                   | 15         |
> | MLA       | 1024         | 23   | 64                   | -          |
> | TPA       | 1024         | 48   | 64                   | -          |
> | SAS       | 1024         | 48   | 96                   | -          |
> | *2.7B parameters* |  |  |  |  |
> | MHA       | 2560         | 32   | 80                   | -          |
> | MQA       | 2560        | 63   | 80                   |  63        |
> | GQA       | 2560         | 62   | 80                   | 31         |
> | MLA       | 2560         | 54   | 80                   | -          |
> | TPA       | 2560         | 96   | 80                   | -          |
> | SAS       | 2560         | 96   | 96                   | -          |
>
> Why do we have such a configuration? Our configuration follows the GPT-3 paper [1] and TPA paper [2].
> * From 125M to 2.7B, the layer number, hidden size, learning rate, and so on follow the GPT-3 paper.
> * We simulate triple attention heads (e.g., 36 simulated heads for 125M with original 12 heads) because the previous model TPA uses the largest attention head number, which is almost 36 heads for 1the 25M model.
>
> For the hidden size (use 125M model as an example):
> * For MHA:
>    * hidden size: 768. This means that the hidden size is 768 for token embedding.
>    * Attention, head number. The 125M has 12 attention heads and 64 hidden size per head, so that 12 x 64=768.
>    * Given a token with embedding 768, we will have query, key, and value, all with shape 768. And then we will reshape them to [12 64] so that we will have 12 attention heads and 64 hidden states per head.
> * For SAS
>    * We will use non-linear mapping to increase the attention head number, usually the triple attention head number, so that we have query/key with shape [36 64].
>    * Also, we will increase the hidden size per head so that we may have [36, 80] for query and key. Therefore, with a very small parameter cost, we simulate more attention heads and hidden size per head.
>
> **Q3: The time cost**
>
> A3: We evaluate the time cost via H100.
>
> *The time cost with different model sizes and micro batch size 1, with H100*
> | Method | 125M  | 350M   | 2.7B    | 6.7B|10.6B|
> |--------|--------|---------|----------|----------|----------|
> | MHA    | 36.20  | 72.87   | 117.82   |186.58|217.01|
> | GQA    | 38.87  | 71.66   | 133.76   |205.69| 239.16|
> | MQA    | 39.37  | 72.01   | 132.62   |202.61| 238.85|
> | MLA    | 51.68  | 104.95  | 180.03   |254.43|637.78(Zero3 to avoid OOM;H200)
> | TPA    | 44.19  | 86.56   | 151.71   |235.85|258.41|
> | SAS    | 68.07  | 138.53  | 215.13   |350.70|402.15|
>
> * The computional cost cost of MHA:$O(12Th^2d^2)+O(T^2hd)$, while $h$ is the attention head number, $d$ is the hidden size per head and $T$ is the sequence length.
> * The computional cost of SAS: $O(11T h^2 d^2+Th'^2d'^2)+O(T^2h'd')$, while $h'$ is the simulated attention head number, $di$ is the simualted hidden size per head and $T$ is the sequence length.
> * Usually, the $h'/h$ is between 1 to 3, and the $d'/d$ is between 1 to 1.5 (less than 1.2 for larger models to control cost)
>
> **Q4: The training lengths of 512 and 1024 are too short. It remains unclear whether SAS maintains its advantages in long-context settings**
>
> A4: We have provided the SAS performance with length 2048 in Section 4.1. And we also conducted an experiment with SAS with a length of 16384. The following are the results.
>
> *The validation perplexity with 125M model size and 16384 training length on the Pile, with H100/200*
> | Model                     | 5K|10K|15K|20K|25K|30K|35K| 40K|45K|50K|
> |-----|-----|-----|-----|-----|-----|-----|-----|-----|-----|-----|
> | MHA                 | 18.31 | 14.50 | 12.85 | 11.96 | 11.45 | 11.11 | 10.44 | 10.37 | 9.95 | 9.94|
> MQA |18.02 | 14.46 | 12.82 | 11.96 | 11.44 | 11.10 | 10.46 | 10.38 | 9.96 | 9.99|
> | GQA            | 17.74 | 14.40 | 12.86 | 11.96 | 11.47 | 11.14 | 10.45 | 10.39 | 9.99 | 9.98|
> |MLA|18.23 | 14.61 | 12.93 | 11.97 | 11.45 | 11.08 | 10.40 | 10.33 | 9.90 | 9.93|
> | SAS         | **14.93** | **12.45** | **11.23** | **10.56** | **10.16** | **9.90** | **9.32** | **9.28** | **8.95** | **8.97**|
>
> **Q5: thorough double-checking of all reported metrics is necessary to ensure accuracy.**
>
> A5: Thank you very much for your notice. We have carefully checked the accuracy and the highlight.
>
> Reference:
>
> [1] Brown, T., Mann, B., Ryder, N., Subbiah, M., Kaplan, J. D., Dhariwal, P., ... & Amodei, D. (2020). Language models are few-shot learners. Advances in neural information processing systems, 33, 1877-1901.
>
> [2] Zhang, Y., Liu, Y., Yuan, H., Qin, Z., Yuan, Y., Gu, Q., & Yao, A. C. (2025). Tensor product attention is all you need. arXiv preprint arXiv:2501.06425.
>
> **If the response addresses the concerns, could you consider improving the score?**

---

> ### Author Response · Authors · 2025-08-04
> **Kind Reminder of Discussion Period**
>
> Dear Reviewer wQTE,
>
> Hope this finds you well.
>
> We have addressed the concerns in the above response, including the discussion of Figure 1, model configuration, time cost, the performance of length 16384, and so on. As the discussion period will be closed soon, if possible, could you consider improving the score?

---

### Official Review · Reviewer_rUv9 · 2025-07-01

**Clarity:** 3
**Significance:** 3
**Originality:** 3
**Rating:** 5
**Confidence:** 3

**Summary:**

The authors propose a novel method of enhancing the performance of transformer language models with small extra cost to the model size, which they call SAS.

The number of attention heads and latent dimensions are 'expanded' by a non-linear transformation, before applying the attention mechanism. After that, the expanded heads are aggregated down to the normal number of heads. This 'simulates' having more attention heads and latent dimensions without increasing the number of parameters.

SAS shows better perplexity on decoder-only transformers compared to MHA, MQA, GQA, MLA, and TPA, and better performance on downstream datasets.

**Questions:**

Did the authors conduct an experiment with larger models, preferably >10B parameters?

At the PEAA step, the authors just average out multiple 'virtual' heads together. In my view, this averaging may cause information loss contained in the virtual heads. Adding another non-linear layer to the $h_i$s, before averaging them may prevent this information loss. Did the authors attempt something like this?

**Ethical Concerns:**

["NO or VERY MINOR ethics concerns only"]

**Final Justification:**

The authors have provided a comparison against MHA at a fixed test time cost. SAS performs on par with MHA with much fewer parameters. This is significant and impressive. So I have raised my score.

**Limitations:**

Yes

**Paper Formatting Concerns:**

On L200, 'Linear_2' is probably meant by the second 'Linear_1'.

**Quality:**

3

**Strengths And Weaknesses:**

**Strengths**

SAS seems to slightly improve the perplexity and downstream performance on 125M - 1.5B parameter models.

The ablation study proves that artifically expanding the number of heads and latent dimensions helps model performance.

**Weaknesses**

While SAS does not increase the number of parameters by much, the increased computational cost may not be worth the slight improvement of model quality. The paper does not state how much the computational cost (e.g. latency or FLOPS) increased, or how it compares to the computational cost of other baselines.

As seen on Figures 2 and 4, it seems that the larger the models gets, the smaller the improvement in perplexity; the gap between the baselines and SAS in the 2.7B model seems to be much smaller than the gap in the 125M model. If this trend continues, it seems possible that SAS might not be very effective at sizes over 10B.

---

> ### Author Rebuttal · Authors · 2025-07-31
>
> Dear Reviewer rUv9,
>
> Thank you very much for the support. We will address the concerns below.
>
> **Q1: The disucssion of computational cost**
>
> A1: We further test the time cost on H100 GPUs.
> *The time cost with different model sizes and micro batch size 1, with H100*
> | Method | 125M  | 350M   | 2.7B    | 6.7B|10.6B|
> |--------|--------|---------|----------|----------|----------|
> | MHA    | 36.20  | 72.87   | 117.82   |186.58|217.01|
> | GQA    | 38.87  | 71.66   | 133.76   |205.69| 239.16|
> | MQA    | 39.37  | 72.01   | 132.62   |202.61| 238.85|
> | MLA    | 51.68  | 104.95  | 180.03   |254.43|637.78(Zero3 to avoid OOM;H200)
> | TPA    | 44.19  | 86.56   | 151.71   |235.85|258.41|
> | SAS    | 68.07  | 138.53  | 215.13   |350.70|402.15|
>
> * The computional cost cost of MHA:$O(12Th^2d^2)+O(T^2hd)$, while $h$ is the attention head number, $d$ is the hidden size per head and $T$ is the sequence length.
> * The computional cost of SAS: $O(11T h^2 d^2+Th'^2d'^2)+O(T^2h'd')$, while $h'$ is the simulated attention head number, $di$ is the simualted hidden size per head and $T$ is the sequence length.
> * Usually, the $h'/h$ is between 1 to 3, and the $d'/d$ is between 1 to 1.5 (less than 1.2 for larger models to control cost)
>
> **Q2: The SAS with larger training (e.g. larger model size)**
>
> A2: The following is the result of SAS with model size 6.7B and 13B on the Pile dataset, which contains both the Arxiv and Books datasets.
>
> *The validation perplexity with 6.7B and 10.6B model size and 512 training length on the Pile, with H100/200*
> | Model                     | 5K|10K|15K|20K|25K|30K|35K| 40K|45K|50K|
> |-----|-----|-----|-----|-----|-----|-----|-----|-----|-----|-----|
> |6.7B|
> | MHA                 | 23.45 | 17.25 | 14.51 | 13.21 | 12.07 | 11.21 | 10.52 | 10.28 | 9.91 | 9.73|
> | MQA |21.56 | 16.32 | 13.93 | 12.73 | 11.70 | 10.90 | 10.24 | 10.02 | 9.66 | 9.49|
> |GQA|21.62 | 16.35 | 13.93 | 12.74 | 11.69 | 10.88 | 10.24 | 10.01 | 9.65 | 9.49|
> |MLA|22.22 | 16.67 | 14.26 | 13.05 | 11.99 | 11.19 | 10.53 | 10.29 | 9.91 | 9.74|
> | SAS         | **20.44** | **15.32** | **13.17** | **12.15** | **11.21** | **10.51** | **9.93** | **9.73** | **9.40** | **9.25**|
> |10.6B|
> | MHA                 | 23.67 | 17.34 | 14.59 | 13.18 | 12.00 | 11.15 | 10.46 | 10.20 | 9.82 | 9.64|
> | MQA |22.13 | 16.55 | 14.03 | 12.80 | 11.70 | 10.88 | 10.21 | 9.98 | 9.61 | 9.43|
> | GQA            | 21.86 | 16.45 | 13.92 | 12.70 | 11.61 | 10.81 | 10.17 | 9.94 | 9.58 | 9.41|
> |MLA|22.40 | 16.67 | 14.22 | 12.99 | 11.90 | 11.11 | 10.42 | 10.18 | 9.83 | 9.66|
> | SAS         | **20.53** | **15.12** | **12.92** | **11.90** | **10.98** | **10.29** | **9.71** | **9.51** | **9.18** | **9.04**|
> * From model size 125M to 10.6B, the SAS always achieves the best performance. And the improvement of SAS in 6.7B and 10.6B is still large enough.
> * The TPA meets overflow performance, and the loss scale is reduced to a minimum of 1 at step 3600 with fp16, and the loss becomes nan with bf16, with a larger model.
>
>
>
>
>
>
>
>
> **Q3: Adding another non-linear layer before averaging them**
>
> A3: The following is the results. The initial embedding is MHA. And the following is the operation (use 125M as example):**SAS-NonLinearBack**:
> * After SAS attetnion, we will have tensor with shape [36 64], while 36 is the head number and 64 is the hidde size.
> * For SAS, we will reshape it to [3 12 64] and then mean pooling to have [1 12 64].
> * For **SAS-NonLinearBack**, we will have a two-layer-gelu (MLP-GeLU-MLP) on the [36 64] map to back to [12 64] (first MLP [36 64] to [36 64]; second MLP [36 64] to [12 64]).
>
> *The perplexity result on the Books dataset with length 512 and H100*
>
> | Model                     | 5K|10K|15K|20K|25K|30K|35K| 40K|45K|50K|
> |-----|-----|-----|-----|-----|-----|-----|-----|-----|-----|-----|
> | MHA                 | 59.23 | 46.31 | 44.68 | 39.33 | 36.57 | 35.44 | 33.19 | 30.35 | 30.29 | 28.68 |
> |SAS-NonLinearBack|54.85 | 43.94 | 42.62 | 37.62 | 34.93 | 34.05 | 31.90 | 29.15 | 29.10 | 27.65 |
> | SAS         | **54.34** | **43.56** | **42.36** | **37.31** | **34.73** | **33.84** | **31.70** | **29.13** | **29.07** | **27.52** |
>
> According to the above results, the mean pooling achieves better performance. This may be caused by the mean pooling is easier for the network to be optimized.
>
> **Q4: On L200, 'Linear_2' is probably meant by the second 'Linear_1'.**
>
> A4: Thank you very much for the notice. We have fixed it in the revision.
>
> **If the response addresses the concerns, could you consider improving the score?**

---

> > ### Comment · Reviewer_rUv9 · 2025-08-05
> > **Response**
> >
> > Thank you for the thorough rebuttal.
> >
> > It seems like the computational cost of SAS is around 1.8x the cost of MHA. I'm curious how much the performance of MHA would improve if the hyperparameters of MHA were adjusted so that its test-time latency matches that of SAS?

---

> ### Author Response · Authors · 2025-08-05
>
> Dear Reviewer rUv9，
>
> Thank you very much for your response. We will answer your question below.
>
> **Q1: The performance of MHA and SAS with comparable time cost**
>
> A1: **We have discussed it in *Response to All Reviewers***. We directly copy the results in the following.
>
> | Model                     |5K|10K|15K|20K|25K|30K|35K| 40K|45K|50K| Time Cost|
> |-----|-----|-----|-----|-----|-----|-----|-----|-----|-----|-----| -----|
> |2.7B|
> |**SAS**| **20.67** | **15.82** | **13.66** | **12.66** | **11.72** | **10.99** | **10.38** | **10.19** | 9.85 | 9.70| 215.13|
> |6.7B|
> | MHA                 | 23.45 | 17.25 | 14.51 | 13.21 | 12.07 | 11.21 | 10.52 | 10.28|9.91 | 9.73|186.58|
> | SAS         | 20.44 | 15.32 |13.17 | 12.15 | 11.21 | 10.51 | 9.93 | 9.73 | 9.40 | 9.25|350.70|
> |10.6B|
> | **MHA**                 | 23.67 | 17.34 | 14.59 | 13.18 | 12.00 | 11.15 | 10.46 | 10.20|**9.82** | **9.64**|217.01|
> | SAS         | 20.53 | 15.12 | 12.92 | 11.90 | 10.98 | 10.29 | 9.71 | 9.51 | 9.18 | 9.04|402.15|
>
> * **With the same model size, the SAS is better than MHA.**.
> * **With the same time cost and smaller model size, SAS 2.7B has comparable performance with MHA 6.7B and MHA 10.6B**.
>
> If there are any other questions, please let us know. **If we have addressed the concerns, could you consider improving the score?**
>
> Again, thank you very much for your attention, and wish you a good day.

---

> > ### Comment · Reviewer_rUv9 · 2025-08-05
> > **Response**
> >
> > Thanks for the comparison. I must have missed the "Response to All Reviewers". This has addressed all my concerns. I will raise my score.

---

> > > ### Author Response · Authors · 2025-08-05
> > >
> > > Dear Reviewer rUv9,
> > >
> > > Thank you very much for the support, which really helps a lot!
> > >
> > > Again, thank you very much for your attention, and wish you a good day.

---

### Official Review · Reviewer_yD2Q · 2025-07-06

**Clarity:** 3
**Significance:** 1
**Originality:** 3
**Rating:** 4
**Confidence:** 3

**Summary:**

This paper suggests simulated attention score as a means to increase the expressivity of the model so as to improve model performance. Given the prior findings that a larger feature dimension as well as the number of heads contribute to better generalization, this method circumvents the bounded number of heads via expanding the dimensions with linear projections. The increased computational costs are compensated by PEAA which applies a grouping for feature aggregation across multiple heads. Results suggest that SAS outperforms baselines, such as MHA, consistently.

**Questions:**

- How does SAS compare with other baselines in terms of compute? A pareto front graph would display the value of the model better.

- Are the baselines state-of-the-art? Judging by the nature of the proposed method, results are largely experimental, which may require well-tuned models for comparison purposes. How different hyperparameters or other transformer techniques may affect the performance is not clear.

**Ethical Concerns:**

["NO or VERY MINOR ethics concerns only"]

**Final Justification:**

Thank you for the clarifications and I understood much better and by seeing other feedbacks to other reviewers, I'm more confident to raise my score.

**Limitations:**

Yes

**Paper Formatting Concerns:**

None.

**Quality:**

2

**Strengths And Weaknesses:**

Strengths:

- This paper has a good clarity, in that the method is built on top of popular MHA and is simple in nature, which is also well presented throughout the paper. I appreciate that the author offers the raw implementation in Appendix H, which helps reproducibility early on.

- Experimental results seem strong and consistently outperforms other attention structures in multiple benchmarks.


Weaknesses:

- The motivation of the method is weak. Discussions on the rationale how the number of heads contribute to better performance are largely omitted.

- Performance gains don’t seem substantial. Different hyperparameters may change the results.

---

> ### Author Rebuttal · Authors · 2025-07-31
>
> Dear Reviewer yD2Q,
>
> Thank you very much for your comment. We will address your concerns below.
>
> **Q1: The motivation of the method is weak. Discussions on the rationale how the number of heads contribute to better performance are largely omitted.**
>
> A1: **We have discussed the effect of the number of attention heads in Figure 1 and Section 4.3 Ablation Study: The Effect of Attention Head Number. For convenience, we copy the result in the following**
>
> According to Figure 1: When only a single attention head (1 head) is used, the model achieves a perplexity score of 6.08. However, as the number of attention heads incrementally increases, the model’s performance improves, reaching an optimal perplexity of 5.82 when the head count matches the original configuration of 12 heads. Beyond this point, further increasing the number of attention heads leads to a gradual degradation in performance, as evidenced by the rising perplexity scores.
> Therefore, more attention heads and larger hidden sizes per head may further improve the performance.
>
> In Figure 6: when the attention number is 12, SAS achieves 22.96 ppl, outperforming MHA.
> This implies that SAS can improve the expressiveness of query, key, and value embeddings, leading to better performance even with the same attention heads number. Moreover, increasing the attention head number from 12 to 36 leads to further performance gains, with SAS achieving a perplexity of 22.14. This demonstrates that SAS benefits from additional attention heads, which enhance the expressiveness of query, key and value embeddings and result in better performance compared to MHA.
>
>
>
>
>
> **Q2: Performance gains don’t seem substantial. Different hyperparameters may change the results.**
>
> A2: **We validate the SAS with different lengths, different datasets, different model sizes, different head numbers, different kernel sizes and so on. And the SAS consistently improves the performance.**
> * **Different Length**. We validate the SAS in Section 4.1, with lengths from 512 to 2048.
> * **Different Dataset**. We validate the SAS with the Arxiv and Books dataset. And we also use the FineWeb-Edu dataset for large-scale training and downstream tasks. Also, we use different downstream evaluation datasets, including ARC-E, ARC-C, Hellaswag, PIQA, ScIQ, SocialIQA and Winograde.
> * * **Different Model Size**. We validate the SAS with sizes 125M, 350M and 2.7B in Section 4.1 and Section 4.2.
> * **Different Head Number**. We validate the SAS with different head number in Section 4.3 with different head number.
> * **Different Kernel Size**. We validate the SAS with different kernel sizes, from 1 to 7.
> * **Different Seed**. We validate the SAS with different random seed in Appendix D.
> * **Different initial attention type**. We try different attention methods as the initial query, key and value embedding. We present the results in Appendix G.
>
> **Therefore, the SAS consistently achieves performance gains, with different lengths, different datasets, different model sizes, different kernel sizes, different seeds, different initial attention types, and so on.**
>
> **Q3: How does SAS compare with other baselines in terms of compute? A pareto front graph would display the value of the model better.**
>
> A3: We test the time cost on H100 GPUs.
>
> *The time cost (ms) with different model sizes and micro batch size 1, with H100*
> | Method | 125M  | 350M   | 2.7B    | 6.7B|10.6B|
> |--------|--------|---------|----------|----------|----------|
> | MHA    | 36.20  | 72.87   | 117.82   |186.58|217.01|
> | GQA    | 38.87  | 71.66   | 133.76   |205.69| 239.16|
> | MQA    | 39.37  | 72.01   | 132.62   |202.61| 238.85|
> | MLA    | 51.68  | 104.95  | 180.03   |254.43|637.78(Zero3 to avoid OOM;H200)
> | TPA    | 44.19  | 86.56   | 151.71   |235.85|258.41|
> | SAS    | 68.07  | 138.53  | 215.13   |350.70|402.15|
>
> * The computional cost cost of MHA:$O(12Th^2d^2)+O(T^2hd)$, while $h$ is the attention head number, $d$ is the hidden size per head and $T$ is the sequence length.
> * The computional cost of SAS: $O(11T h^2 d^2+Th'^2d'^2)+O(T^2h'd')$, while $h'$ is the simulated attention head number, $di$ is the simualted hidden size per head and $T$ is the sequence length.
> * Usually, the $h'/h$ is between 1 to 3, and the $d'/d$ is between 1 to 1.5 (less than 1.2 for larger models to control cost)
>
>
>
> **Q4: Are the baselines state-of-the-art?**
>
> A4: **The baseline is the state-of-the-art. The baselines contain the most recent works, including MLA [1] and TPA[2].**
>
> **Q5: How different hyperparameters or other transformer techniques may affect the performance is not clear.**
>
> A5: We have proved in Q2 that SAS consistently achieves performance gains, with different lengths, different datasets, different model sizes, different kernel sizes, different seeds, different initial attention types and so on. We further validate the model siz, the larger model size and training length.
>
> *The following is the performance with 6.7B and 10.6B.*
> | Model                     |5K|10K|15K|20K|25K|30K|35K| 40K|45K|50K|
> |-----|-----|-----|-----|-----|-----|-----|-----|-----|-----|-----|
> |6.7B|
> | MHA                 | 23.45 | 17.25 | 14.51 | 13.21 | 12.07 | 11.21 | 10.52 | 10.28 | 9.91 | 9.73|
> | MQA |21.56 | 16.32 | 13.93 | 12.73 | 11.70 | 10.90 | 10.24 | 10.02 | 9.66 | 9.49|
> |GQA|21.62 | 16.35 | 13.93 | 12.74 | 11.69 | 10.88 | 10.24 | 10.01 | 9.65 | 9.49|
> |MLA| 22.22 | 16.67 | 14.26 | 13.05 | 11.99 | 11.19 | 10.53 | 10.29 | 9.91 | 9.74|
> | SAS         | **20.44** | **15.32** | **13.17** | **12.15** | **11.21** | **10.51** | **9.93** | **9.73** | **9.40** | **9.25**|
> |10.6B|
> | MHA                 | 23.67 | 17.34 | 14.59 | 13.18 | 12.00 | 11.15 | 10.46 | 10.20 | 9.82 | 9.64|
> | MQA |22.13 | 16.55 | 14.03 | 12.80 | 11.70 | 10.88 | 10.21 | 9.98 | 9.61 | 9.43|
> | GQA            | 21.86 | 16.45 | 13.92 | 12.70 | 11.61 | 10.81 | 10.17 | 9.94 | 9.58 | 9.41|
> |MLA|22.40 | 16.67 | 14.22 | 12.99 | 11.90 | 11.11 | 10.42 | 10.18 | 9.83 | 9.66|
> | SAS         | **20.53** | **15.12** | **12.92** | **11.90** | **10.98** | **10.29** | **9.71** | **9.51** | **9.18** | **9.04**|
>
> *The validation perplexity with 125M model size and 16384 training length on the Pile, with H100/200*
> | Model                     | 5K|10K|15K|20K|25K|30K|35K| 40K|45K|50K|
> |-----|-----|-----|-----|-----|-----|-----|-----|-----|-----|-----|
> | MHA                 | 18.31 | 14.50 | 12.85 | 11.96 | 11.45 | 11.11 | 10.44 | 10.37 | 9.95 | 9.94|
> MQA |18.02 | 14.46 | 12.82 | 11.96 | 11.44 | 11.10 | 10.46 | 10.38 | 9.96 | 9.99|
> | GQA            | 17.74 | 14.40 | 12.86 | 11.96 | 11.47 | 11.14 | 10.45 | 10.39 | 9.99 | 9.98|
> |MLA|18.23 | 14.61 | 12.93 | 11.97 | 11.45 | 11.08 | 10.40 | 10.33 | 9.90 | 9.93|
> | SAS         | **14.93** | **12.45** | **11.23** | **10.56** | **10.16** | **9.90** | **9.32** | **9.28** | **8.95** | **8.97I
>
> **If the response addresses the concerns, could you consider improving the score?**
>
> Reference:
>
> [1] Liu, A., Feng, B., Wang, B., Wang, B., Liu, B., Zhao, C., ... & Xu, Z. (2024). Deepseek-v2: A strong, economical, and efficient mixture-of-experts language model. arXiv preprint arXiv:2405.04434.
>
> [2] Zhang, Y., Liu, Y., Yuan, H., Qin, Z., Yuan, Y., Gu, Q., & Yao, A. C. (2025). Tensor product attention is all you need. arXiv preprint arXiv:2501.06425.

---

> > ### Comment · Reviewer_yD2Q · 2025-08-07
> >
> > Although the performance gain is modest relative to the added computational cost, it is consistently observed across all datasets. This finding offers a valuable contribution to the machine learning community, as the impact of the number of attention heads is an under-explored topic in the current literature. These results can provide practical guidance for selecting an appropriate number of heads. This is of growing importance as models continue to scale.
> >
> > However, the reason behind the better performances in larger number of headcount is not so obvious. The authors suggest consistent empirical evidence in multiple figures, but essentially, what it means by the "expressiveness" of an increasing headcount, and its possible interpretation effort is missing. As such, we cannot answer questions as to what is the maximum number of heads (in the void of any specific metric or markers/footprints) or what is the differential inner-workings of the attention layer characterized by each feature. Therefore, I tend toward keeping the original rating.

---

> ### Author Response · Authors · 2025-08-04
> **Kind Reminder of Discussion Period**
>
> Dear Reviewer yD2Q,
>
> Hope this finds you well.
>
> We have addressed the concerns in the above response, including time cost, the effect of attention head number, different hyperparameters, and so on. As the discussion period will be closed soon, if possible, could you consider improving the score?

---

> ### Author Response · Authors · 2025-08-06
> **Kind Reminder of Discussion Period Ends**
>
> Dear Reviewer yD2Q,
>
> We have addressed the concerns in the above response, including time cost, the effect of attention head number, different hyperparameters, and so on. If there are any questions, please let us know before the discussion period ends (August 8 AoE). After the detailed explanation, if possible, could you consider improving the score?
>
> Again, thank you for your attention, and wish you a good day.

---

> ### Author Response · Authors · 2025-08-07
> **The Theoritical Analysis**
>
> Dear Reviewer yD2Q,
>
> Thank you very much for your response. We will answer your question below.
>
> **Q1: The Theoretical Analysis of maximum attention head number**
>
> A1: **The meaning of increasing the head number to improve expressiveness: output relation pattern independently. The answer for the maximum number of heads: $O(D^2)$.**
>
> The Softmax Attention with Fully Parameterized Bilinear Attention [1,2].
>
> $O_i = \sum_{c=1}^{H}\left(\sum_{j=1}^i\phi(\frac{x_{i}W_{c}x_{j}}{\sqrt{D}})x_jU_c\right)$
>
> where:
> $W_c \in \mathbb{R}^{D \times D}$ = learnable bilinear transformation for head $c$,
> $\phi$ = the kernel function such as softmax,
> $U_c \in \mathbb{R}^{D \times D}$ = value projection matrix,
> $H$ = number of attention heads,
> $D$ = hidden dimension per head.
>
> Each head computes a weighted sum of values, with weights determined by $x_i W_c x_j^T$. **For $ m$-dimensional $x_i$ and $ n$-dimensional $x_j$, one potential relation pattern is related to $x_i^mx_j^n$. Therefore, at most $D^2$ relations or interactions**).
>
>
> The bilinear term $x_i W_c x_j^T=\sum_{m=1}^D \sum_{n=1}^D x_i^{(m)} W_c^{(m,n)} x_j^{(n)}$ can be expressed as $(x_i \otimes x_j) \cdot \text{vec}(W_c)$, where $\text{vec}(W_c) \in \mathbb{R}^{D^2}$.
> With $H$ heads, the model can represent up to $D^2$ distinct relations [6]. **We hope that the output of each head is a single relation but not mixed relations, so that we can utilize the relations independently (e.g., $\phi$ processes the relation independently).**
>
>
> **Example**
> * Assume $Rank(W_c)=1$, and there is only one non-zero value in $W_c$. And there are $H=D^2=4$.
> * Input：
> ```math
> x_i = [a b]
> x_j =[c d]
> ```
> * $W_c$:
> ```math
> W_1= [1 0]
>        [0 0]
> W_2= [0 1]
>        [0 0]
> W_3= [0 0]
>        [1 0]
> W_4= [0 0]
>        [0 1]
> ```
> The first head $W_1$ will gives the result of relation $ac$, the $W_2$, $W_3$, $W_4$ will give the result of $ad$, $bc$, $bd$. Therefore, this ($ac$, $ad$, $bc$, $bd$) suggests different attention patterns and relations. **And they could be adjusted independently and not mixed (e.g., $\phi$ processes the relation independently), compared to only one head with $W_c=[ 1~ 1 $ \\ \\ $ 1   ~1]$ that outputs the mix of $ac+ad+bc+bd$.** **With a large $D$, we can have more combinations. With a large $H$, we can capture more independent relations. And previous [5, 6, 7] suggest that the expressiveness is related to O(HD), where $H$ is the attention head and $D$ is the hidden size per head.**
>
>
> If we increase $D$ (e.g., hidden size per head), we increase the maximum number of relations. If we increase the $H$, we increase the number of independent relations that are actually captured.
>
> And this is why GQA, MQA, MLA, and TPA have the potential to increase performance, though they are low-rank: they either increase the attention head number (GQA, MQA, MLA, and TPA) or increase the hidden size per head (MLA), with the same parameter cost.
>
> **If the response addresses the concerns, could you consider improving the score?**
>
> Reference:
>
> [1] Vaswani, A., Shazeer, N., Parmar, N., Uszkoreit, J., Jones, L., Gomez, A. N., ... & Polosukhin, I. (2017). Attention is all you need. Advances in neural information processing systems, 30.
>
> [2] Shazeer, N. (2019). Fast transformer decoding: One write-head is all you need. arXiv preprint arXiv:1911.02150.
>
> [3] Shazeer, N., Lan, Z., Cheng, Y., Ding, N., & Hou, L. (2020). Talking-heads attention. arXiv preprint arXiv:2003.02436.
>
> [4] Ainslie, J., Lee-Thorp, J., De Jong, M., Zemlyanskiy, Y., Lebrón, F., & Sanghai, S. (2023). Gqa: Training generalized multi-query transformer models from multi-head checkpoints. arXiv preprint arXiv:2305.13245.
>
> [5] Liu, A., Feng, B., Wang, B., Wang, B., Liu, B., Zhao, C., ... & Xu, Z. (2024). Deepseek-v2: A strong, economical, and efficient mixture-of-experts language model. arXiv preprint arXiv:2405.04434.
>
> [6] Hu, J., Li, H., Zhang, Y., Wang, Z., Zhou, S., Zhang, X., ... & Jiang, D. (2024). Multi-matrix factorization attention. arXiv preprint arXiv:2412.19255.
>
> [7] Zhang, Y., Liu, Y., Yuan, H., Qin, Z., Yuan, Y., Gu, Q., & Yao, A. C. (2025). Tensor product attention is all you need. arXiv preprint arXiv:2501.06425.

---

> > ### Author Response · Authors · 2025-08-07
> >
> > Dear Reviewer yD2Q,
> >
> > Thank you for your attention and precious time. **In the above response, we have answered the question: 1) the "expressiveness" of an increasing headcount; 2) the maximum number of heads.**
> > * **For Detailed Response**: Please refer to the above response.
> > * **For a summary of the Response**
> >    *  The meaning of increasing the head number to improve expressiveness: capture the relation pattern independently.
> >    * The answer for the maximum number of heads: $O(D^2)$.
> >    * Previous work also suggests that expressiveness is related to $O(HD)$, where $H$ is the head number and $D$ is the hidden size per head [1,2,3].
> >
> > Again, thank you very much for your attention, and wish you a good day.
> >
> >
> > Reference:
> >
> >
> > [1] Liu, A., Feng, B., Wang, B., Wang, B., Liu, B., Zhao, C., ... & Xu, Z. (2024). Deepseek-v2: A strong, economical, and efficient mixture-of-experts language model. arXiv preprint arXiv:2405.04434.
> >
> > [2] Hu, J., Li, H., Zhang, Y., Wang, Z., Zhou, S., Zhang, X., ... & Jiang, D. (2024). Multi-matrix factorization attention. arXiv preprint arXiv:2412.19255.
> >
> > [3] Zhang, Y., Liu, Y., Yuan, H., Qin, Z., Yuan, Y., Gu, Q., & Yao, A. C. (2025). Tensor product attention is all you need. arXiv preprint arXiv:2501.06425.

---

> > > ### Author Response · Authors · 2025-08-08
> > > **Discussion Will End In One Day**
> > >
> > > Dear Reviewer yD2Q,
> > >
> > > Thank you for your attention and precious time. In the above response, we have answered the question: 1) the "expressiveness" of an increasing headcount; 2) the maximum number of heads. As the discussion will end in one day, could we know whether there are any remaining concerns? If the above response addresses the concerns, could you improve the score?

---

> > > > ### Author Response · Authors · 2025-08-09
> > > > **Discussion Will End In 8 hours**
> > > >
> > > > Dear Reviewer yD2Q,
> > > >
> > > > Thank you for your attention and precious time. In the above response, we have answered the question: 1) the "expressiveness" of an increasing headcount; 2) the maximum number of heads. As the discussion will end in 8 hours, could we know whether there are any remaining concerns? Again, thank you for your attention.

---

### Author Response · Authors · 2025-08-01
**Response to All Reviewers**

Dear All Reviewers,

Thank you for the comment for suggestion. We have finished the specific question of reviewers. In the following, we will further clarify the common questions. We will really appreciate it if Reviewer yD2Q, Reviewer wQTE and Reviewer DaRN could consider improving the score, as the main concerns are well-addressed.

**The main target of this work: The main target of this work is improving the performance with the same/comparable parameter cost**. And the following are the two main conclusions.
* **With the same parameter cost, the SAS is better than all other methods, including MHA, MQA, GQA, MLA, TPA, and so on.**
* **With a few parameters, the SAS has the potential to be better than other methods.** For example, 2.7B SAS achieves comparable performance with MHA 10.6B at the final stage, while their time cost is comparable.

**Q1: The Time Cost (Reviewer yD2Q, Reviewer rUv9, Reviewer wQTE, Reviewer DaRN)**

A1: The following are the results.

*The time cost with different model sizes and micro batch size 1, with H100*
| Method | 125M  | 350M   | 2.7B    | 6.7B|10.6B|
|--------|--------|---------|----------|----------|----------|
| MHA    | 36.20  | 72.87   | 117.82   |186.58|217.01|
| GQA    | 38.87  | 71.66   | 133.76   |205.69| 239.16|
| MQA    | 39.37  | 72.01   | 132.62   |202.61| 238.85|
| MLA    | 51.68  | 104.95  | 180.03   |254.43|637.78(Zero3 to avoid OOM;H200)
| TPA    | 44.19  | 86.56   | 151.71   |235.85|258.41|
| SAS    | 68.07  | 138.53  | 215.13   |350.70|402.15|


* The computional cost cost of MHA:$O(12Th^2d^2)+O(T^2hd)$, while $h$ is the attention head number, $d$ is the hidden size per head and  $T$ is the sequence length.
* The computional cost of SAS: $O(11T h^2 d^2+Th'^2d'^2)+O(T^2h'd')$,  while $h'$ is the simualted attention head number, $d'$ is the simulated hidden size per head and  $T$ is the sequence length.
* Usually, the $h'/h$ is between 1 to 3, and the $d'/d$ is between 1 to 1.5 (less than 1.2 for larger models to control cost).


**Q2: The SAS with larger model size and training length (Reviewer yD2Q, Reviewer rUv9, Reviewer wQTE)**

A2: The following is the result of SAS with a 2.7B to 6.7B model size and training length 16384 on Pile dataset.



*The following is the performance and clocking time with 6.7B and 10.6B.*
| Model                     |5K|10K|15K|20K|25K|30K|35K| 40K|45K|50K|
|-----|-----|-----|-----|-----|-----|-----|-----|-----|-----|-----|
|2.7B|
|SAS| 20.67 | 15.82 | 13.66 | 12.66 | 11.72 | 10.99 | 10.38 | 10.19 | 9.85 | 9.70|
|6.7B|
| MHA                 | 23.45 | 17.25 | 14.51 | 13.21 | 12.07 | 11.21 | 10.52 | 10.28 | 9.91 | 9.73|
| MQA |21.56 | 16.32 | 13.93 | 12.73 | 11.70 | 10.90 | 10.24 | 10.02 | 9.66 | 9.49|
|GQA|21.62 | 16.35 | 13.93 | 12.74 | 11.69 | 10.88 | 10.24 | 10.01 | 9.65 | 9.49|
|MLA|54.34 | 22.22 | 16.67 | 14.26 | 13.05 | 11.99 | 11.19 | 10.53 | 10.29 | 9.91 | 9.74|
| SAS         | **20.44** | **15.32** | **13.17** | **12.15** | **11.21** | **10.51** | **9.93** | **9.73** | **9.40** | **9.25**|
|10.6B|
| MHA                 | 23.67 | 17.34 | 14.59 | 13.18 | 12.00 | 11.15 | 10.46 | 10.20 | 9.82 | 9.64|
| MQA |22.13 | 16.55 | 14.03 | 12.80 | 11.70 | 10.88 | 10.21 | 9.98 | 9.61 | 9.43|
| GQA            | 21.86 | 16.45 | 13.92 | 12.70 | 11.61 | 10.81 | 10.17 | 9.94 | 9.58 | 9.41|
|MLA|22.40 | 16.67 | 14.22 | 12.99 | 11.90 | 11.11 | 10.42 | 10.18 | 9.83 | 9.66|
| SAS         | **20.53** | **15.12** | **12.92** | **11.90** | **10.98** | **10.29** | **9.71** | **9.51** | **9.18** | **9.04**|

The validation perplexity with 125M model size and 16384 training length on the Pile, with H100/200*
| Model                     | 5K|10K|15K|20K|25K|30K|35K| 40K|45K|50K|
|-----|-----|-----|-----|-----|-----|-----|-----|-----|-----|-----|
| MHA                 | 18.31 | 14.50 | 12.85 | 11.96 | 11.45 | 11.11 | 10.44 | 10.37 | 9.95 | 9.94|
MQA |18.02 | 14.46 | 12.82 | 11.96 | 11.44 | 11.10 | 10.46 | 10.38 | 9.96 | 9.99|
| GQA            | 17.74 | 14.40 | 12.86 | 11.96 | 11.47 | 11.14 | 10.45 | 10.39 | 9.99 | 9.98|
|MLA|18.23 | 14.61 | 12.93 | 11.97 | 11.45 | 11.08 | 10.40 | 10.33 | 9.90 | 9.93|
| SAS         | **14.93** | **12.45** | **11.23** | **10.56** | **10.16** | **9.90** | **9.32** | **9.28** | **8.95** | **8.97**|

* **With the same parameter cost, the SAS is definitely better than all other methods, validated by model size from 125M to 10.6B and training length 512 to 16384.**.
* Even with a few parameters, the SAS has the potential to be comparable with all other methods. For example, SAS 2.7B achieves a comparable performance to baseline MHA 10.6B.

---

### Author Response · Authors · 2025-08-06
**The Core Contribution**

Dear Area Chair and Reviewer,

Thank you for the engagement in the discussion. **We are grateful for the support of the Reviewer rUv9 and Reviewer wQTE to improve the score, and we wonder whether Reviewer yD2Q and Reviewer DaRN could reconsider the decision as the concerns are addressed.**

**The main contribution of this work**
* We propose the concept of simulated head and simulated hidden size per head. By such a concept, we could simulate a large attention head number and hidden size per head at a smaller cost.
* We validate the proposed method via different training lengths (from 512 to 16384), different model sizes (from 125M to 10.6B), different kernel sizes (from 1 to 7), different datasets (Arxib, Books, Pile, and FineWeb-Edu), different random seeds, and so on.
* SAS 2.7B could achieve comparable performance with MHA 10.6B.

We believe that the SAS could bring a new concept to the attention and model. And the work could inspire the following model architecture.


Again, thank you for your attention, and wish you a good day

---

### Note · Authors · 2025-08-12

Dear Area Chair and Reviewer,

Thank you for the attention and support. We explain the core contribution of this work further.

**The core contribution of this work**

* We propose the concept of simulated attention head and hidden size per head.
* **The Core Claim (Agreed By All Reviewers): with the same parameter cost, the SAS is definitely better than all other methods .**
* The 2.7B SAS has the potential to achieve comparable performance with 10.6B MHA.
* We validate the SAS with different lengths (from 512 to 16384), different model sizes (from 125m to 10.6B), and different datasets (Arxiv, Books, Pile, and FinWeb-Edu
* **We provide theoretical analysis about the model performance with different head and hidden size per head for Fully Parameterized Bilinear Attention (FPBA), where the hidden size per head is equal to  hidden size**
   * Recently work proposes to use more attention head or hidden size per head to improve performance.  However, it is not clear how they affect the performance. **For example, GLM 4.5 (8 Aug, 2025) [1] doubles attention head to have better result, but no explanation. However, based on our analysis, it is possible to explain why the model is better with double attention head.**
   * According to our analysis, $m$-th element of $x_i$ and $ n$-th  $x_j$, one potential relation pattern is related to $x_i^mx_j^n$. Therefore, at most $D^2$ relations or interactions, while $D$ is the hidden size per head for FPBA.
   * If we increase $D$, we increase the maximum number of relations. If we increase $H$, we increase the number of independent relations that are captured. If increase hidden size per head $d$, we reduce the low-rank mapping information loss (e.g. 2-d [a b] map to 1d [a+b] have information loss).
   * For detailed analysis, we could refer to response *"The Theoritical Analysis" for **Fully Parameterized Bilinear Attention (FPBA)***
   * **For FPBA, the $D=d$ is the hidden size per head/hidden size. For non-FPBA, such as the MHA, the $D$ is the hidden size.**

**We believe this work not only gives a better attention mechanism, but also helps understand the attention works. We truly believe that this work will inspire and guide future research, enabling others to develop even more effective attention-based methods.**

Again, thank you for the attention and support, and wish you a good day.

Reference:

[1] GLM Team. (2025). GLM-4.5: Agentic, Reasoning, and Coding (ARC) Foundation Models. arXiv preprint arXiv:2508.06471.

---

### Decision · Program_Chairs · 2025-09-17

**Decision:**

Accept (poster)

**Comment:**

The paper develops a novel concept of simulated attention head and simulated hidden size per head. This concept should translate to performance lifts with smaller models and make them as powerful as larger models. This is a very interesting idea and there was consensus about this. However, the major concern was the overhead of training such models might outweigh the advantage of training  smaller models. This trade off seems to be not clear. During post rebuttal discussion the following points emerged.

<em>For the benchmarks given by the authors, the reduction in perplexity seems to be equivalent to non-SAS with about 20K additional training steps -- in general, as SAS seems to add 2x-4x overheard per training step, this is generally not a favorable trade-off</em>

<em>The authors give one example in which SAS performs better than non-SAS at equivalent wall-clock time: it is the 2.7B SAS model compared to the larger (10B) MHA model. This is a good example, but the larger models for this training setup don't seem to provide much benefit over the smaller ones. That is, one can always use a model that is too big in order to provide an example with very high wall-clock time and low quality, and this may be the situation</em>

Given these concerns, it is thus not absolutely clear on how much impact the paper will have on the practice of large transformers and hence it is difficult to make a case of strong acceptance. Nevertheless,
the paper already brings fresh ideas which should be of interest to Neurips community. If accepted,  the author(s) should  modify the manuscript with all the additional experiments discussed in rebuttal and incorporate appropriate commentary in the manuscript to address the concerns listed above.